# Constraining tectonic uplift and advection from the main drainage divide of a mountain belt

Chuanqi He [1,2], Ci-Jian Yang [2], Jens M. Turowski[2], Gang Rao [1✉], Duna C. Roda-Boluda [2] &
Xiao-Ping Yuan[2]

One of the most conspicuous features of a mountain belt is the main drainage divide. Divide location is influenced by a number of parameters, including tectonic uplift and horizontal advection. Thus, the topography of mountain belts can be used as an archive to extract tectonic information. Here we combine numerical landscape evolution modelling and analytical solutions to demonstrate that mountain asymmetry, determined by the location of the main drainage divide, increases with increasing uplift gradient and advection velocity. Then, we provide a conceptual framework to constrain the present or previous tectonic uplift and advection of a mountain belt from the location and migration direction of its main drainage divide. Furthermore, we apply our model to Wula Shan horst, Northeastern Sicily, and Southern Taiwan.

[1] Key Laboratory of Geoscience Big Data and Deep Resource of Zhejiang Province, School of Earth Sciences, Zhejiang University, 310027 Hangzhou, China.
[2] German Research Centre for Geosciences (GFZ), 14473 Potsdam, Germany. ✉email: raogang@zju.edu.cn

Mountain belts provide natural boundaries on the Earth's surface. They influence atmospheric circulation[1,2], determining regional weather and the distribution of hydrological systems[3,4], and limit the migration of plants and animals, affecting biodiversity[3,5]. Mountain building is driven by tectonic forces (vertical uplift and horizontal advection, Fig. 1) and erosion[6–12]. Consequently, many geohazards such as earthquakes and landslides are concentrated in mountainous regions[13,14]. Systematic constraints on the spatiotemporal patterns of tectonic deformation help to improve the understanding of the interactions among lithosphere, hydrosphere, biosphere, and atmosphere. Various methods have been used to constrain patterns of tectonic deformation. For example, over short timescales of years to decades, a combination of global positioning system (GPS)-measured horizontal velocities and Interferometric Synthetic Aperture Radar (InSAR) measurements can provide high-accuracy regional uplift rates[15–18]. The average vertical slip rate of a fault can be obtained by offset measurements combined with precise dating of displaced materials[19,20]. Longer-term exhumation and uplift history can be estimated by low-temperature thermochronometry[21,22], inversion of river long profiles using topographic data[23,24], radiogenic isotope dating of lava flows[25], and combination of seismic reflection and other geological data (e.g., drill core)[26].

Here, we present a new method to provide constraints on the tectonic pattern and history of a mountain belt solely based on the location and mobility of its main drainage divide (MDD). First, we quantify the relationship between mountain asymmetry, determined by MDD location, and both uplift gradient and advection velocity through theoretical reasoning and landscape evolution modelling. Then, we illustrate the interaction and competition between tectonic deformation and erosion in determining MDD mobility. Collectively, we build a model to provide constraints on the present or previous uplift and horizontal advection of a mountain belt based on the current location and mobility of its MDD, and apply this model to Wula Shan horst, Northeastern Sicily, and Southern Taiwan. Specifically, we demonstrate that: (1) the difference in uplift rate between the two edges of Wula Shan horst is 0.14 mm/year; (2) the tectonic activity of Northeastern Sicily remains constant or has slightly changed; (3) the tectonic activity of Southern Taiwan may have increased, and we expect that under the persistence of present conditions, the main drainage divide of Southern Taiwan will migrate southeastwards.

## Results

**Relationship between uplift gradient and divide location.** For landscape evolution simulations and analytical solution (see "Methods"), we consider a one-dimensional (1D) asymmetric mountain belt within an uplift field that linearly increases from one edge to the other, with an uplift gradient $\lambda$ (mm/year per km). Normalised divide location ($d$) is defined as the ratio between the distance from the MDD to the lower uplifting edge (or the width of the positive side, Fig. 1) and the width of the mountain belt. From an initial random topography without MDD, a mountain belt evolves towards steady state (uplift rate = erosion rate) with a divide position dependent on uplift gradient (Fig. 2a–d, Movie 1). For a spatially uniform uplift rate of 0.5 mm/year, the MDD forms near the centre ($d = 49.7 \pm 3.5\%$) of the mountain range (Fig. 2a), consistent with previous works[6,10,27]. With a linear asymmetric uplift of 0.5 mm/year at the bottom edge and values from 1.0 mm/year ($\lambda = 0.01$ mm/year per km) to 10 mm/year ($\lambda = 0.19$ mm/year per km) at the top edge, the MDD migrates closer to the top boundary (Fig. 2b–d). Consequently, the normalised divide location increases from $d = 57.6 \pm 4.1\%$ to a maximum of $d = 73.3 \pm 2.7\%$ (Fig. 2e). Based on the stream power model, we obtain an analytical solution of the divide location, which is independent of erodibility (see "Methods"). The analytical solution generally agrees with the numerical results, and falls within their 95% confidence range (Fig. 2e). However, for the same uplift gradient, the analytical solution predicts higher $d$ values than the results of numerical models. This may be because the Hack's law[28] exponent $b$ used for the

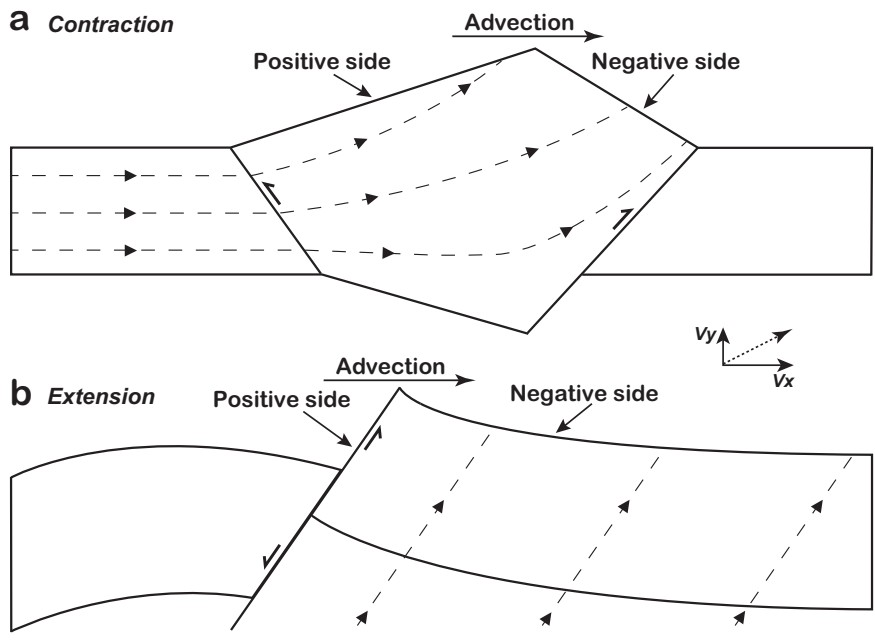

**Fig. 1 Kinematic models for tectonic uplift and horizontal advection.** Material transport (marked as dashed lines with arrows), with vertical component (Vx) and horizontal component (Vy), causes tectonic uplift and horizontal advection in convergent orogen **a** (modified after refs. [6, 34]) and extensional orogen **b**. For each system, advection is from the positive side to the negative side. For the positive side, due to advection, the range half-width tends to increase. By contrast, the range half-width of the negative side tends to decrease.

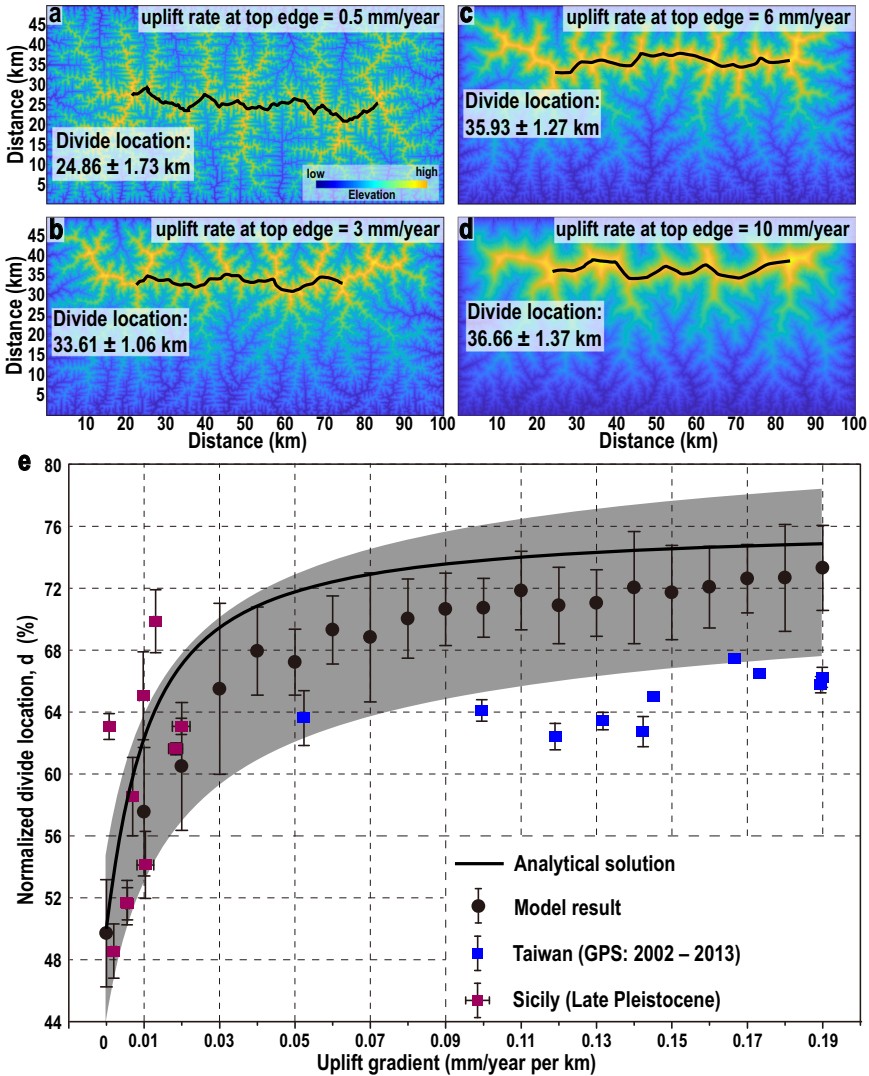

**Fig. 2 Relationship between uplift gradient and normalised divide location. a–d** Selected model results of divide position in response to asymmetric uplift. Topographies of asymmetric mountain range generated by the TopoToolbox Landscape Evolution Model (TTLEM[40]) in response to a linear gradient in uplift rate from 0.5 mm/year at the bottom edge to **a** 0.5 mm/ year, **b** 3 mm/year, **c** 6 mm/year, and **d** 10 mm/ year at the top edge. Black lines are the main drainage divides of mountain ranges. A run time of 300 Myr (million years) is sufficient to attain steady state from the initial topography in all cases (Supplementary Fig. 6). **e** Relationship between uplift gradient and steady-state divide location. The data points denote the mean values with one standard deviation. Each data point of Southern Taiwan and Northeastern Sicily is calculated from three values (see "Methods"). For numerical simulation, 2000–3000 values are used to calculate divide location, depending on the length of the main drainage divide. Grey shading represents the 95% confidence range of the model results. Uplift rates of Northeastern Sicily integrated since the Late Pleistocene (~125,000 years) are calculated from data in ref. [47]. Uplift rates of Southern Taiwan are calculated from decadal-scale GPS data. Source data are provided as a Source Data file.

analytical solution ($b = 1.6$, see "Methods") does not fit exactly with the simulations[27]. In numerical models, the $b$ values vary with uplift gradient (Supplementary Figs. 1–4). The analytical solution demonstrates that $d$ generally increases with increasing $b$ (Supplementary Fig. 5).

**Relationship between advection velocity and divide location.** Generally, there are multiple possible situations for mountain belt kinematics. Here, we consider two simple parameterisations of mountain belt kinematics with advection[6,27]. First, advection velocity is spatially uniform, hence, the only variable is velocity. Second, advection velocity decreases linearly from the positive side to the negative side (Fig. 1), corresponding to a constant rate of shortening[6]. In this case, two independent parameters determine the distribution of advection: the velocity gradient and the velocity at the edge of the negative side. We obtain the analytical solutions

for both cases (Fig. 3a). When advection velocity is spatially uniform, $d$ increases linearly with velocity. When it is zero (i.e., no advection), the MDD forms at the centre of the mountain belt. When velocity reaches about 5.1 mm/year, $d$ is 100%, namely, there is no MDD within the topography. The analytical solution shows that the velocity at which $d$ reaches 100% depends on erodibility, mountain width, and hillslope length (see "Methods"). In the case of a constant rate of shortening, $d$ increases with both velocity gradient and the velocity at the edge of the negative side, in agreement with previous works[6,8,27].

There are two first-order scenarios when a mountain belt experiences uplift gradient and advection simultaneously, which we explore via three other numerical simulations (Fig. 3b–e). First, the positive side has a lower uplift rate than the negative side. In this case, both uplift and advection push the MDD towards the negative side (as in the case of Southern Taiwan, see

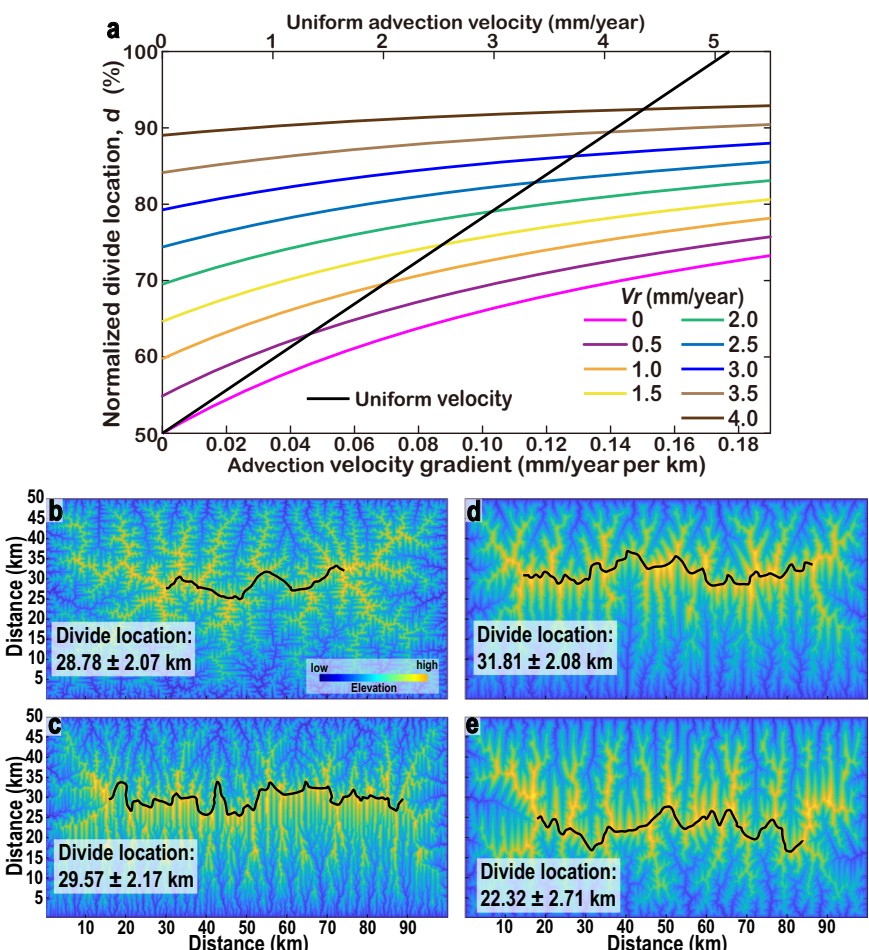

**Fig. 3 Divide response to advection. a** Relationship between advection velocity and normalised divide location. $V_r$ is the velocity at the edge of the negative side. **b** No advection. Uplift rate increases linearly from the bottom edge of 0.5 mm/year to 1 mm/year at the top edge, with an uplift gradient of 0.01 mm/year per km. This simulation belongs to the 20 numerical models in Fig. 2e. **c** Uniform uplift rate of 0.5 mm/year. Advection towards the top side, and advection velocity decreases linearly from the bottom edge of 1 mm/year to 0.5 mm/year at the top edge, with a velocity gradient of 0.01 mm/year per km. **d** Uplift gradient and advection push the divide in the same direction. Uplift rate increases linearly from the bottom edge of 0.5 mm/year to 1 mm/year at the top edge, with an uplift gradient of 0.01 mm/year per km. Advection towards the top side, and advection velocity decreases linearly from the bottom edge of 1 mm/year to 0.5 mm/year at the top edge, with a velocity gradient of 0.01 mm/year per km. **e** Uplift gradient and advection push the divide towards different directions. Uplift rate increases linearly from the bottom edge of 0.5 mm/year to 1 mm/year at the top edge, with an uplift gradient of 0.01 mm/year per km. Advection towards the bottom side, and advection velocity decreases linearly from the top edge of 1 mm/year to 0.5 mm/year at the bottom edge, with a velocity gradient of 0.01 mm/year per km.

discussion below). Second, the positive side has a higher uplift rate than the negative side. In this case, uplift gradient and advection push the MDD towards different directions (as in Northeastern Sicily, see discussion below). When uplift gradient and advection push the MDD in the same direction, the mountain belt becomes more asymmetric (Fig. 3b–d). In contrast, when they push the divide in different directions, their effects partially cancel out. In this case, the steady-state divide location is dominated by the driver with a higher contribution to divide migration, and the mountain belt is less asymmetric (Fig. 3b, c, e). Generally, the time needed to reach steady state decreases with increasing uplift gradient (Supplementary Fig. 6).

## Discussion

**Deriving tectonic information from the MDD.** The MDD tends to migrate towards the side with a higher uplift rate[10,27,29,30], lower precipitation and/or higher rock resistance[10,27,29,31–33], and in the direction of advection[6,8,27,32,34,35]. Divide dynamics can be

understood conceptually by focusing on three forces: advection, asymmetric uplift, and asymmetric erosion. Erosion is controlled by tectonic deformation, which can start or increase instantaneously, but erosion itself takes some time to adjust to those changes. This leads to some temporary decoupling between tectonic forcing and erosion. Hence, during transient periods, one of these factors may be the dominant force controlling divide mobility. Without advection, the steady-state MDD is located in a position determined by uplift gradient (Fig. 2). If uplift gradient is zero (i.e., uniform uplift), the steady-state MDD is at the centre of the mountain belt, and the erosion rates are equal on both sides. Any asymmetry in divide location generates an asymmetry in slope and thus in erosion rate, leading to divide migration towards the centre of the range. Conceptually, when asymmetric uplift outweighs erosional contrast in influencing divide mobility, the MDD moves to the side with higher uplift rate (Fig. 4a). This further steepens the catchments on the side with higher uplift rate, and consequently, increases the erosional contrast between the two sides. This leads to an enhanced influence of the erosional

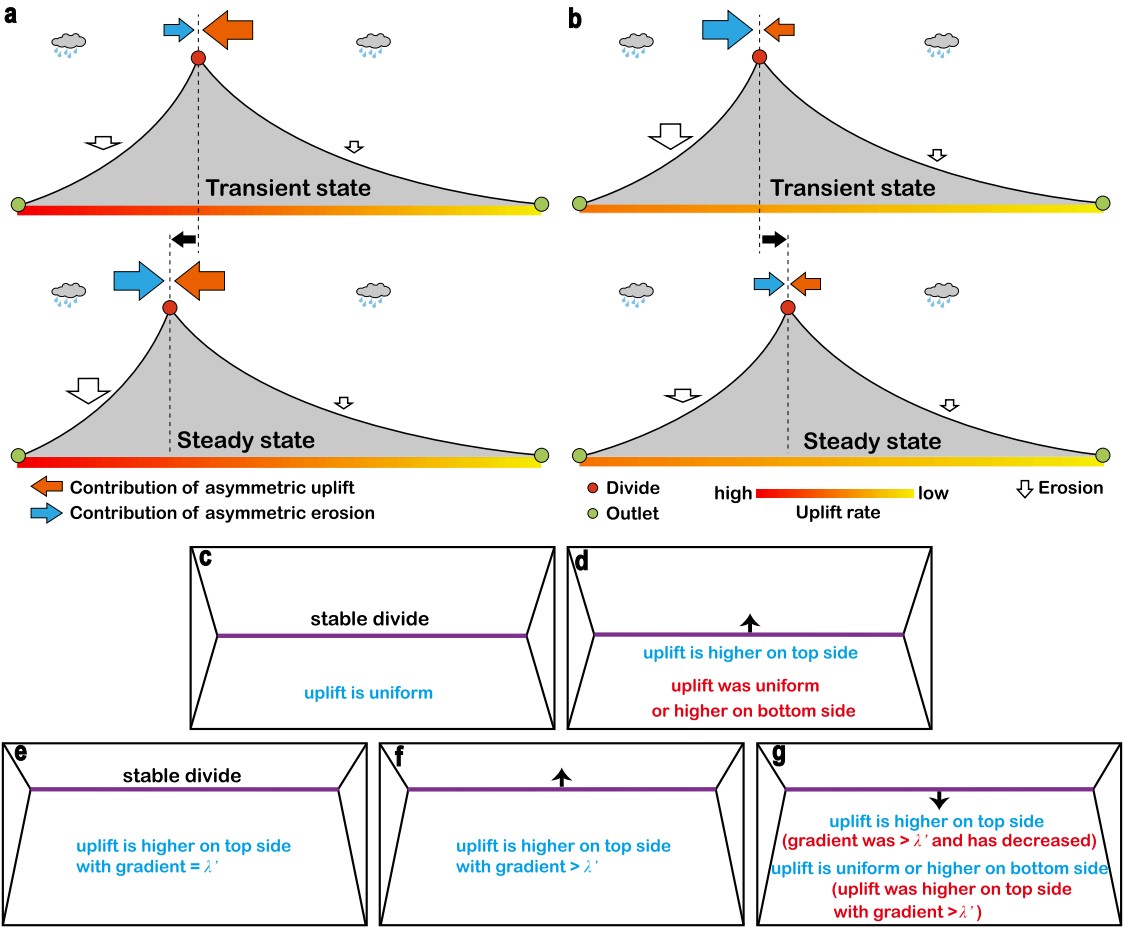

**Fig. 4 Divide migration in response to asymmetric uplift and erosion, and its implications for constraining tectonic information. a** When asymmetric uplift dominates divide mobility, divide moves to the side with a higher uplift rate until it reaches steady state. **b** When erosional contrast predominantly influences divide mobility, divide migrates to the side with a lower erosion rate to reach equilibrium. **c** Symmetric mountain belt with a stable divide. **d** Symmetric mountain belt with an unstable divide. **e** Asymmetric mountain belt with a stable divide. **f** Asymmetric mountain belt with a divide that is migrating to the steeper side. g, Asymmetric mountain belt with a divide that is moving to the gentler-sloping side. Purple lines are the divide, with its direction of motion marked by arrows. Constraints on present uplift and its history are given in the blue and red comments, respectively.

contrast across the MDD in controlling divide mobility until the contributions from asymmetric uplift and erosional contrast become equal at steady state. Alternatively, when erosional contrast outweighs asymmetric uplift in controlling divide mobility, the MDD moves towards the side with lower erosion rate (Fig. 4b). As a consequence, the erosional contrast decreases, diminishing its contribution to divide mobility until the contributions of asymmetric uplift and erosional contrast become equal at steady state. The interaction and competition between advection and erosion in controlling divide mobility are similar to the pattern of uplift gradient and erosion (Supplementary Fig. 7).

The relationship between $d$ and $\lambda$ (Fig. 2), together with the conceptual framework (Fig. 4a, b), allow us to derive tectonic information from the location and mobility of the MDD. Both of these parameters can easily be obtained from present topography[33]. If the controls of precipitation, lithology, and advection on divide location can be ruled out as dominant, we can distinguish five possible cases. For a symmetric mountain belt, the uplift is uniform if the MDD is stable (Fig. 4c). When the MDD is migrating towards the top side, the uplift is higher on the top side, and was uniform or higher on the bottom side (Fig. 4d). For an asymmetric mountain belt, we can apply the theoretical relationship (Fig. 2e) to acquire a reference uplift gradient $\lambda'$ from the current divide location. If the MDD is stable, the uplift is higher on the steeper side with a gradient of $\lambda'$ (Fig. 4e). When the MDD

of an asymmetric mountain belt is unstable, $\lambda'$ can provide a lower bound on the present or previous gradient. If the MDD is migrating towards the steeper side, the uplift is higher on the steeper side with a gradient greater than $\lambda'$ (Fig. 4f). When the MDD is moving to the gentler-sloping (bottom) side, the uplift was higher on the top side with a gradient greater than $\lambda'$ (Fig. 4g). Additionally, if the present uplift is known to be higher on the top side, the uplift gradient should have decreased.

In principle, the current divide location and migration pattern can also be used to constrain advection pattern (Supplementary Fig. 7). However, there is no known way of establishing divide stability in the case with advection. Thus, our model is unable to derive advection information quantitatively until a method that could establish divide mobility under advection conditions is developed. Nevertheless, if the present advection velocity is known, our model can provide information on previous tectonic deformation, depending on the specific uplift and advection patterns (see case studies below).

**Application to natural landscapes**. For the Wula Shan horst in northern China, which may have negligible advection, the uplift rate is higher at the southern edge and decreases northwards[30]. The present MDD is stable, with a normalised divide location of 60.8% (Supplementary Fig. 8), placing it in the case described in Fig. 4e. Thus, its present uplift gradient is estimated to be

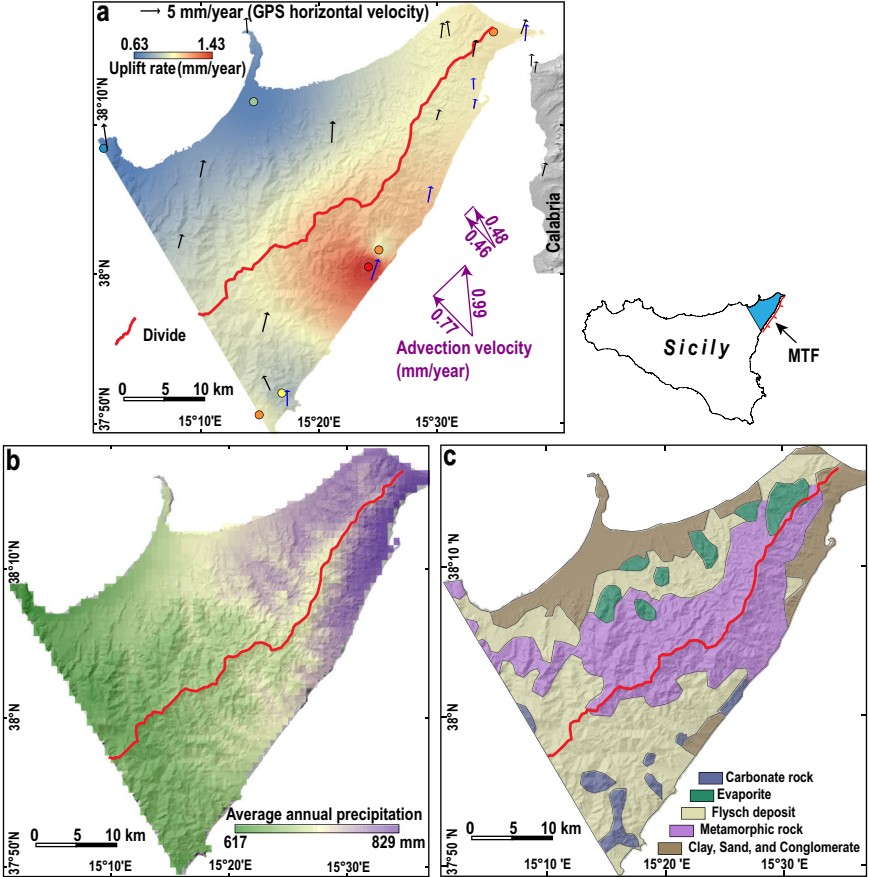

**Fig. 5 Natural example of Northeastern Sicily. a** Uplift rates integrated since the Late Pleistocene (data from ref. [47]). Black and blue arrows indicate GPS horizontal velocities in a fixed central Europe frame[48]. See "Methods" for the details on the estimation of advection velocity. **b** Average annual precipitation between 1970 and 2000, acquired from WorldClim[49]. **c** Lithology of Northeastern Sicily (modified after ref. [50]). MTF: Messina-Taormina Fault (modified after ref. [45]).

0.008 mm/year per km using Eq. (10). As the mountain width is 17.5 km, the difference in uplift rate between the southern and northern edges is estimated to be 0.14 mm/year. Thus, if precise constraints on the uplift rate related to the southern piedmont fault with pronounced fault scarps[30] are available, the uplift rate at the northern edge could also be estimated. The Wula Shan horst is a simple example. In more complex cases, when the present uplift gradient and advection velocity are known, our model can provide constraints on previous tectonic deformation. We discuss two further examples: Northeastern Sicily and Southern Taiwan.

In Northeastern Sicily, uplift is mainly controlled by the tectonic activity of the Messina-Taormina Fault (MTF, a normal fault), being higher on the SE side (Fig. 5a). The estimated average uplift gradient of 0.0092 mm/year per km (see "Methods") pushes the MDD towards the SE. In the direction perpendicular to the MDD, the average advection velocity is estimated to be 0.46 or 0.77 mm/year (Fig. 5a) (see "Methods"). The northwestward advection pushes the MDD towards the NW. Both the mean annual precipitation (711 and 705 mm/year for the NW and SE flanks, respectively) and lithology are nearly symmetrical with respect to the MDD (Fig. 5b, c), and thus should have little influence on divide location. We choose the ratio between the width of the NW side and the mountain width as the normalised divide location, $d$. Although the $d$-$\lambda$ data points generally follow the trend of the analytical solution for mountain belts with uplift gradient, most points fall below the analytical solution due to the existence of advection (Fig. 2e). We numerically solve the divide

location under different settings including both uplift gradient and advection (Supplementary Fig. 9, see Code availability), where no analytical solution is possible. For an advection velocity of 0.46 mm/year, the MDD will form at $d = 56.7\%$ under the present settings, comparable with the current divide location of 58.7%, implying that the tectonic activity remains constant over the response timescale of the MDD. This value only slightly decreases to $d = 56.4\%$ if we exclude the precipitation difference between the two sides of the range. By contrast, if we exclude uplift gradient and advection, the MDD is predicted at $d = 49.4\%$ and 62.2%, respectively. Alternatively, for an advection velocity of 0.77 mm/year, the MDD should form at $d = 52.8\%$ under the present settings, lower than the current divide location. Based on our model, we attribute this discrepancy to the decreased uplift gradient and/or increased advection velocity. In summary, we speculate that the tectonic activity of Northeastern Sicily remains constant or has slightly changed.

For the orogenic wedge of Southern Taiwan, the GPS-derived uplift gradient of 0.14 mm/year per km pushes the MDD towards the SE (Fig. 6a). Owing to advection, rock moves from the NW to the SE through the mountain belt coming to the surface near the Longitudinal Valley (Fig. 6b)[6]. Accordingly, we choose the Longitudinal Valley as the reference frame for advection. Other points within the mountain belt are moving towards the SE with respect to the valley. This advection, with a velocity gradient of 0.35 mm/year per km (see "Methods"), also pushes the MDD towards the SE[6]. The average annual precipitations of 2528 and 2231 mm/year for the NW and SE flanks, respectively (Fig. 6b),

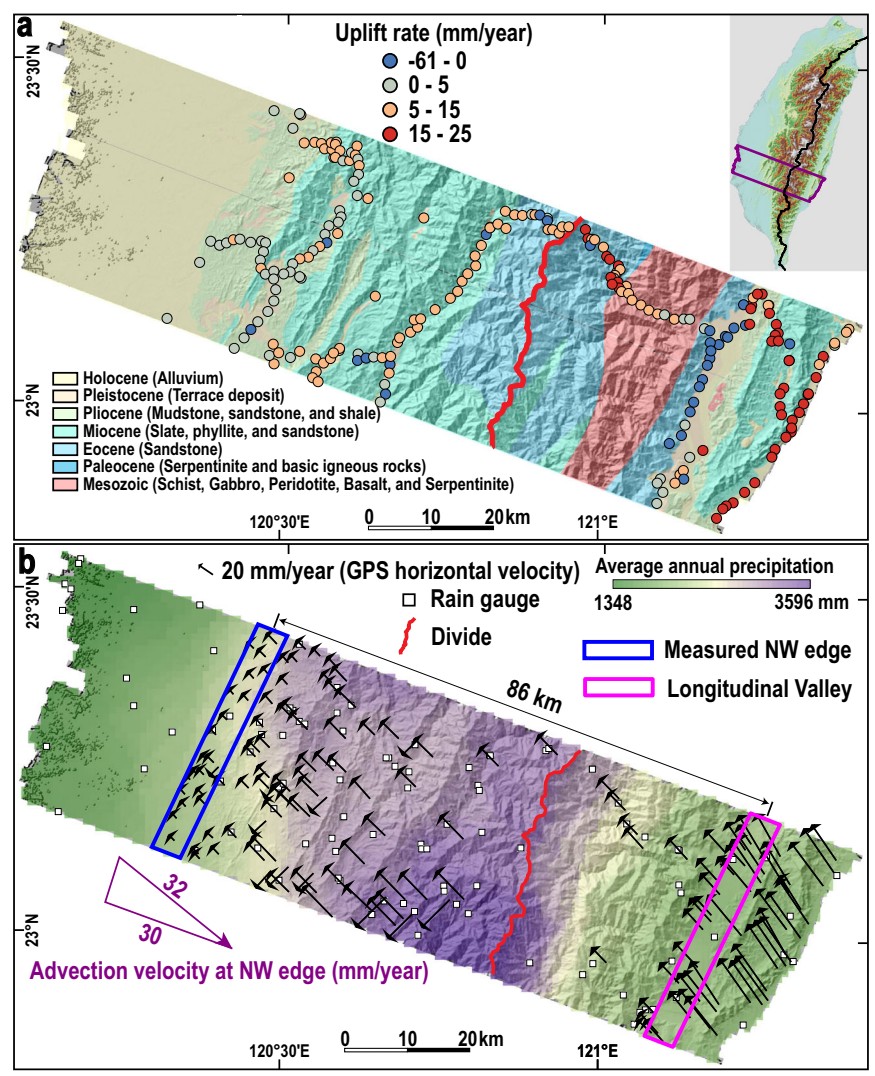

**Fig. 6 Natural example of Southern Taiwan. a** GPS-derived uplift rates (2002–2013) and lithology of Southern Taiwan. **b** GPS horizontal velocities in a fixed Eurasia frame and average annual precipitation of Southern Taiwan (1990–2010). See "Methods" for the details on the estimation of advection velocity.

create an erosion contrast, pushing the MDD towards the SE. Thus, uplift gradient, advection, and precipitation all push the MDD towards the SE. Although the $d$-$\lambda$ data points generally follow the trend of the model results and analytical solution, the $d$ values are still lower than the model prediction including uplift gradient only (Fig. 2e). Under the present settings, at steady state, the divide location is estimated to be at $d = 78.1\%$ (Supplementary Fig. 10), which is much higher than the current value of 64.7%. If we exclude the precipitation difference, the divide location will change slightly to $d = 76.6\%$. Under the present tectonic settings, the mountain belt becomes symmetric at $d = 49.2\%$ only after imposing a tremendous precipitation difference (2 m/year and 8 m/year on the NW and SE sides, respectively). Thus, tectonic deformation primarily determines the divide location of Southern Taiwan, while precipitation is a secondary control. A similar feature has been observed in Southern Alps of New Zealand, where the uplift gradient and advection have pushed the MDD close to the NW coastline, despite precipitation rates of ~12 m/year and 1 m/year on the NW and SE sides, respectively, which tend to push the MDD towards the SE[27]. We attribute the discrepancy between the current divide location of $d = 64.7\%$ and the predicted value of $d = 78.1\%$ to two possible reasons. One is that the uplift rates and advection velocity

estimated from GPS observations over several decades may have been over-estimated due to the fact that the study area displayed a high level of seismicity[36] and hence is dominated by large interseismic elastic accumulation. The other possible reason is a recent increase in tectonic activity, which is likely due to the southward propagation of the arc-continent collision in Taiwan[34,37–39]. Such a condition implies that the mountain belt is in a transient state with respect to its divide position. We can expect that under the persistence of present conditions, the MDD of Southern Taiwan will migrate southeastwards, approaching a divide location at $d = 78.1\%$.

In simulations and analytical considerations, we assume a linear uplift gradient, as this requires fewer constraints and yields results that are readily compared to the existing numerical studies[10,29]. Assuming a nonlinear uplift field described by a power law with exponent $\alpha$ ($\alpha = 1$ is a linear uplift gradient; $\alpha \neq 1$ is a nonlinear uplift gradient), mountain asymmetry increases with increasing $\alpha$ (Supplementary Fig. 11). As a result, for nonlinear uplift gradients, the conceptual relationships described in Fig. 4 can be used to constrain uplift patterns rather than provide a quantitative uplift gradient value.

Our calculations and concepts offer an opportunity to obtain first-order constraints on the present tectonic uplift and advection

of a mountain belt and its history via simple observations of freely available remote-sensing data. As such, this approach does not require excessive investments of time and money, and can, in principle, be widely applied. Although the method does not yield complete information, it may be a useful supplement to existing, more labour-intensive methods.

## Methods

**Landscape evolution model.** We conduct 23 numerical simulations (Figs. 2, 3b–e) using the TopoToolbox Landscape Evolution Model (TTLEM[40]), a MATLAB-based landscape evolution model contained in TopoToolbox 2[41]. All models consisted of a $50 \times 100$ km rectangular domain, with a spatial resolution of 75 m, in which four edges are fixed to a constant elevation of 0 m. Other simulation parameters are set as follows: erodibility ($k_1 = 3 \times 10^{-6}$/year, exponents of the stream power model $m = 0.5$ for discharge and $n = 1$ for slope, hillslope diffusivity $= 0.03$ m$^2$/year, and drainage area threshold $= 0.2$ km$^2$. We run the models for 300 Myr, with time steps of 0.1 Myr. The steady-state divide location is analysed based on the topographies of the models at 300 Myr.

We estimate the time needed to reach steady state using the e-folding time. The mean elevation of the models increases to a maximum and then decreases to be relatively stable after reaching steady-state (Supplementary Fig. 6). We use $y = a + b \times e^{-t/c}$ to fit the elevation data from the highest point to 300 Myr, where $y$ is the mean elevation, $t$ is time, $a$ and $b$ are constants, $e$ is the Euler number, and $c$ is the e-folding time. The steady-state time is estimated as four times of the e-folding time plus the time used to reach the highest elevation. Hack's parameters ($k_2$ and $b$) are calculated within the TopoToolbox 2[41].

**Analytical solution for uplift gradient.** At steady state, according to the detachment-limited stream power model:[42]

$$S = \left(\frac{U}{k_1}\right)^{\frac{1}{n}} A^{-\frac{m}{n}}, \tag{1}$$

where $S$ is the gradient of river channel, $U$ is uplift rate, $k_1$ is erodibility, $A$ is the upstream drainage area, and $m$ and $n$ are positive constants. According to Hack's law that describes the relationship between length of river channel and drainage area:[28]

$$A = k_2 x^b, \tag{2}$$

where $x$ is the distance from the MDD, $k_2$ and $b$ are constants. We assume that the mountain width is $M$, and the uplift rate of the lower uplift edge is $U_1$, which linearly increases to the higher uplift edge with the uplift rate gradient $\lambda$. Thus, the uplift rate at the lower uplift side is

$$U = U_1 + \lambda(D - x), \tag{3}$$

and the uplift rate at the higher uplift side is

$$U = U_1 + \lambda(D + x), \tag{4}$$

where $D$ is the distance between the MDD and the lower uplift edge. For the lower uplift side, the elevation at the MDD is

$$H = \int_0^D \left(\frac{U_1 + \lambda(D - x)}{k_1}\right)^{\frac{1}{n}} (k_2 x^b)^{-\frac{m}{n}} dx, \tag{5}$$

similarly, the elevation at the MDD for the higher uplift side is

$$H = \int_0^{M-D} \left(\frac{U_1 + \lambda(D + x)}{k_1}\right)^{\frac{1}{n}} (k_2 x^b)^{-\frac{m}{n}} dx. \tag{6}$$

Assuming $m = 0.5$ and $n = 1$, we then obtain the elevation at the lower uplift side as

$$\begin{aligned} H &= \frac{1}{k_1} k_2^{-\frac{1}{2}} \int_0^D (U_1 + \lambda(D - x)) x^{-\frac{b}{2}} dx \\ &= \frac{1}{k_1} k_2^{-\frac{1}{2}} \left(\frac{2U_1}{2-b} D^{\frac{2-b}{2}} + \frac{4\lambda}{(2-b)(4-b)} D^{\frac{4-b}{2}}\right), \end{aligned} \tag{7}$$

and the elevation at the higher uplift side as

$$\begin{aligned} H &= \frac{1}{k_1} k_2^{-\frac{1}{2}} \int_0^{M-D} (U_1 + \lambda(D + x)) x^{-\frac{b}{2}} dx \\ &= \frac{1}{k_1} k_2^{-\frac{1}{2}} \left(\frac{2(U_1 + \lambda M)}{2-b} (M - D)^{\frac{2-b}{2}} - \frac{4\lambda}{(2-b)(4-b)} (M - D)^{\frac{4-b}{2}}\right). \end{aligned} \tag{8}$$

At the MDD, the elevations on both sides are the same, i.e., Eqs. (7) and (8) are equal. In our numerical models, assuming $U_1 = 0.5$ mm/year and $M = 50$ km, we have

$$D^{\frac{2-b}{2}} + \frac{4\lambda}{4-b} D^{\frac{4-b}{2}} = (1 + 100\lambda)(50 - D)^{\frac{2-b}{2}} - \frac{4\lambda}{4-b} (50 - D)^{\frac{4-b}{2}}. \tag{9}$$

The average value of $b$ for the 20 numerical models is 1.6 (Supplementary Fig. 1). There is no systematic correlation between uplift gradient and Hack's parameters (Supplementary Figs. 1–4), which is supported by the observations from natural

landscapes that exhibit self-similarity of drainage basins[43]. Using $b = 1.6$ in Eq. (9), we have

$$D^{0.2}\left(1 + \frac{\lambda D}{0.6}\right) = (50 - D)^{0.2}\left(1 + 100\lambda - \frac{\lambda(50 - D)}{0.6}\right). \tag{10}$$

We plot the relationship between $\lambda$ and normalised divide location, $d$ ($d = D/M$) (Fig. 2e) in MATLAB based on Eq. (10).

**Analytical solution for advection.** The advection theory is based on refs. [4,27].

The base level of the positive side and the negative side are located at $x = D$ and $x = M - D$, respectively. We use $m = 0.5$, $n = 1$, and $b = 2$, as there is no close-form solution for other values of $b$[27]. The steady-state slope for the positive side can be expressed as

$$S = -\frac{U}{k_1 x + V}, \tag{11}$$

and for the negative side,

$$S = \frac{U}{V - k_1 x}, \tag{12}$$

where $S < 0$, $U$ and $V$ are uplift rate and advection velocity, respectively.

We consider two cases for mountain belts with advection. One is the advection velocity linearly decreases from the edge of the positive side to the edge of the negative side. In the other case, advection velocity is spatially uniform. For the first case, at the edge of the positive side, the advection velocity is $V_p$, which linearly decreases to $V_r$ at the boundary of the negative side, with an advection velocity gradient of $\gamma = (V_p - V_r)/M$.

The velocity on the positive side is

$$V = V_p - \gamma(D - x), \tag{13}$$

and the velocity on the negative side is

$$V = V_p - \gamma(D + x). \tag{14}$$

Combining the above equations, we obtain the slope of the positive side as

$$S = -\frac{U}{k_1 x + V_p - \gamma(D - x)}, \tag{15}$$

and the slope of the negative side as

$$S = \frac{U}{V_p - \gamma(D + x) - k_1 x}, \tag{16}$$

The boundary condition for the positive side is $H(x = D) = 0$, and for the negative side is $H(x = M - D) = 0$.

Integrating Eqs. (15) and (16) along the river profile, we can obtain the elevation of the positive side as

$$H = \frac{-U}{k_1 + \gamma} \ln\left(\frac{(k_1 + \gamma)x + V_p - \gamma D}{(k_1 + \gamma)D + V_p - \gamma D}\right), \tag{17}$$

and the elevation for the negative side as

$$H = \frac{-U}{k_1 + \gamma} \ln\left(\frac{(k_1 + \gamma)x + \gamma D - V_p}{(k_1 + \gamma)(M - D) + \gamma D - V_p}\right). \tag{18}$$

At the boundary between hillslope and channel ($x = Xc$), the two sides of the mountain belt have the same elevation, thus,

$$\frac{(k_1 + \gamma)Xc + V_p - \gamma D}{(k_1 + \gamma)D + V_p - \gamma D} = \frac{(k_1 + \gamma)Xc + \gamma D - V_p}{(k_1 + \gamma)(M - D) + \gamma D - V_p}. \tag{19}$$

Therefore, we can calculate the divide location as

$$D = \frac{k_1 M Xc + \gamma M Xc - 2(V_r + \gamma M)Xc + (V_r + \gamma M)M}{2k_1 Xc + \gamma M}. \tag{20}$$

For a spatially uniform velocity ($V$), i.e., $V_p = V_r$, the divide location is

$$D = \frac{M}{2} + \frac{M - 2Xc}{2k_1 Xc} V. \tag{21}$$

We plot the relationship between $d$ and advection velocity (Fig. 3a) based on Eqs. (20) and (21) using $M = 50$ km, $k_1 = 3 \times 10^{-6}$/year, and $Xc = 1.6$ km.

**Data collection and processing for natural landscapes**

*Northeastern Sicily, Italy.* Northeastern Sicily is a ~50-km long NE-SW trending peninsula (Fig. 5a). The SE flank (average slope $= 18.3°$) is steeper than the NW flank (average slope $= 14.6°$). Since the Early-Middle Pleistocene, Northeastern Sicily has been affected by strong uplift driven by extensional faults[44], especially the MTF (Fig. 5a), which actively deforms the Late Quaternary marine terraces and Holocene coastal notches[45,46].

The map of uplift rate integrated since the Late Pleistocene (Fig. 5a) is interpolated using seven uplift data points from ref. [47]. Uplift gradients vary along the MDD, thus, we divide the whole 50-km-wide natural landscape into 30 sub-

swaths that are perpendicular to the MDD. The uplift rate differences between the SE and NW edges for each sub-swath are divided by the length of the sub-swath to obtain $\lambda$. Then, the distance between the NW edge and the MDD is measured as $D$. The ratio between $D$ and mountain width ($M$) is $d$. Values of $\lambda$ and $d$ for every three sub-swaths are averaged as the values of the 5-km-wide main swaths. Therefore, we have ten main swaths with $\lambda$ and $d$ values (Fig. 2e). The MDDs of Wula Shan horst, Northeastern Sicily, and Southern Taiwan are extracted using Advanced Spaceborne Thermal Emission and Reflection Radiometer Global Digital Elevation Model (ASTER GDEM) data with 30 m spatial resolution.

Given the proximity of the SE coast to the MTF that is nearly fixed with respect to the hanging wall (the Calabria, Fig. 5a), the advection velocity is close to the horizontal slip rate on the MTF. Thus, we choose Western Calabria as the reference frame for advection. We calculate the average GPS velocity from three GPS data points along the coast of Western Calabria, and the average GPS velocity from eighteen GPS data points in Northeastern Sicily. The results show that the average advection velocity in Northeastern Sicily is 0.99 mm/year, from SE to NW (Fig. 5a). In the direction perpendicular to the MDD, the advection velocity is 0.77 mm/year. Advection could also be defined as the rock velocity with respect to the erosional boundary. Therefore, we use the SE coastline of Northeastern Sicily as reference frame for advection. We calculate the average GPS velocity from six GPS data points (Fig. 5a, blue arrows) along the SE coastline, and the average GPS velocity from the rest twelve GPS data points in Northeastern Sicily. The results show that the average advection velocity in Northeastern Sicily is 0.48 mm/year (Fig. 5a). In the direction perpendicular to the MDD, the advection velocity is 0.46 mm/year. Collectively, we consider that the advection velocity in Northeastern Sicily is 0.46 or 0.77 mm/year (Fig. 5a).

For the NW side, the Hack's parameters are 0.35 and 1.65 for $k_2$ and $b$, respectively. For the SE side, the Hack's parameters are 0.24 and 1.88 for $k_2$ and $b$, respectively (Supplementary Fig. 12). The mountain width of Northeastern Sicily is 40 km. The hillslope length (value of Xc) is measured as the straight-line distance between the MDD and the locations where channels start to form. We measure 34 sites within Google Earth and obtained an average Xc 1756 ± 294 m. The average channel steepness ($ks$) values are 143 m and 179 m for the NW and SE sides, respectively (see "Data availability"), calculated within TopoToolbox 2[41]. We estimate erodibility using the constraints imposed by $ks$ in the following way. In the first-level iteration, picking a trial value for the divide location, we integrate the gradient function to generate river long profiles for both sides of the mountain belt. We then fit a power law with a fixed exponent of $-0.5$ to the long profile by using a nonlinear least-square fitting algorithm, to obtain a value for $ks$. Erodibility is then iteratively adjusted until the best-fit slope-area relationship matched the observed $ks$. In a second-level iteration, the trial value for the divide position is adjusted until it matched the position calculated using the erodibilies obtained in the first-level iteration. We obtain $k_1$ values of $5.4 \times 10^{-6}$/year and $4.9 \times 10^{-6}$/year for the NW and SE sides, respectively.

We write a MATLAB script (divide_location.m, see Code availability) that numerically integrates the gradient function in the stream power model to estimate the steady-state location. Uplift gradient, advection velocity, Hack's parameters, hillslope length, erodibility, precipitation, and mountain width are required by the script. Our analytical solutions, i.e., Eqs. (10), (20), and (21), are independent of Hack's coefficient, $k_2$, while the script needs $k_2$ as input. The results show that the analytical solutions and the script match best with a $k_2$ value of 1 (Supplementary Fig. 13). Thus, we use a $k_2$ value of 1 for both Northeastern Sicily and Southern Taiwan as the input of the script. Using the script and the above parameters, we estimate the divide location under different settings (Supplementary Fig. 9).

*Southern Taiwan*. As an active collision zone, the Taiwan Orogen formed from the convergence between the Philippine Sea Plate and the Eurasian Plate that started a few million years ago and has continued to the present[10,37]. A 40-km-wide zone is selected for this study, because the uplift rate in this zone linearly increases from the NW edge to the SE edge (Fig. 6a). The SE flank (average slope = 21.5°) is steeper than the NW flank (average slope = 15.0°).

Similar to the method used for Northeastern Sicily, the 40-km-wide area is cut into 30 sub-swaths and 10 main swaths that are perpendicular to the MDD. The uplift rate differences between the southeasternmost and northwesternmost data points for each sub-swath are divided by their distance to calculate $\lambda$. Then, the distance between the NW edge and the MDD is measured as $D$. The ratio between $D$ and mountain width ($M$) is $d$. Values of $\lambda$ and $d$ for every three sub-swaths are averaged as the value of the 4-km-wide main swath. Thus, we have ten main swaths with $\lambda$ and $d$ values (Fig. 2e).

The advection velocity gradient is measured between the NW mountain front and the Longitudinal Valley (Fig. 6b), and the latter is used as the reference frame of advection. We calculate the average advection velocity from the data near the NW mountain front. The results show that the advection velocity in this area is 32 mm/year, and 30 mm/year in the direction perpendicular to the MDD (Fig. 6b). The average advection velocity gradient for Southern Taiwan (0.35 mm/year per km) is obtained as the ratio of advection velocity (30 mm/year) and the distance (86 km) between the measured NW edge and the Longitudinal Valley.

For the NW side, the Hack's parameters are 0.48 and 1.66 for $k_2$ and $b$, respectively. For the SE side, the Hack's parameters are 0.38 and 1.77 for $k_2$ and $b$, respectively (Supplementary Fig. 12). The mountain width of Southern Taiwan is 130 km. We measure 46 sites within Google Earth and obtained an average Xc of

1556 ± 240 m. The average channel steepness is 197 m and 237 m for the NW and SE sides, respectively, calculated within TopoToolbox 2[41]. Similar to the method used for Northeastern Sicily, in Southern Taiwan, we obtain $k_1$ values of $5.8 \times 10^{-5}$/year and $4.1 \times 10^{-5}$/year for the NW and SE sides, respectively. Similarly, with the script and the above parameters, we estimate the divide location under different settings (Supplementary Fig. 10).

## Data availability

Topographies of all the numerical models at 300 Myr and Supplementary Movie 1 are deposited at https://doi.org/10.6084/m9.figshare.12318965.v10. Lithology and GPS-derived uplift rates of Southern Taiwan are acquired from https://www.moeacgs.gov.tw. For Southern Taiwan, GPS horizontal velocities in a fixed Eurasia frame are obtained from https://www.moi.gov.tw. Average annual precipitation of Southern Taiwan is downloaded from https://www.cwb.gov.tw. The 30 m spatial resolution digital elevation model data are obtained from http://www.gscloud.cn. The boundary of Sicily is downloaded from https://map.igismap.com/gis-data, under open database license www.opendatacommons.org/licenses/odbl. Other relevant data supporting the findings of the study are available in the Supplementary Information, or from the corresponding author upon request. Source data are provided with this paper.

## Code availability

A MATLAB script (divide_location.m) to generate a divide location under different settings is deposited at https://doi.org/10.6084/m9.figshare.12318965.v10.

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

## Acknowledgements

We express our gratitude to Niels Hovius, Brian J. Yanites, Wolfgang Schwanghart, Rong Yang, Liran Goren, and Mimmo Palano for fruitful discussions that greatly improved this work. Special thanks are also given to Taylor Schildgen, Joel Scheingross, Mitchell D'Arcy, and Stefano Luigi Gariano for advice on choosing natural landscapes as case studies. This work was supported by the Basic Science Centre Programme for Multiphase Media Evolution in Hypergravity of the National Natural Science Foundation of China (No. 51988101), the National Natural Science Foundation of China (Nos. 41502203 and 41941016), the Second Tibetan Plateau Scientific Expedition and Research (No. 2019QZKK0708), and the National Science and Technology Major Project (No. 2017ZX05008-001).

## Author contributions

C.H., C.-J.Y., and J.T. contributed equally to this work. C.H. and G.R. conceived the idea. G.R. and C.H. directed the project. C.-J.Y. and C.H. constructed the numerical models. J.T., C.H., and X.Y. derived the analytical solutions. C.H., C.-J.Y., D.R.-B., and J.T. contributed to data collection and processing of the natural landscapes. All authors contributed to the interpretation of the data. C.H. and J.T. wrote the original manuscript, and all authors participated in paper revisions.

## Competing interests

The authors declare no competing interests.
