## [Peer Review File · Nature Communications]

Reviewers' comments:

Reviewer #1 (Remarks to the Author):

This paper presents numerical and analytical evidence that, all else equal, the position and mobility of the main divide of a mountain belt should be directly related to the uplift field across the range. The analysis presented is elegant and, in the places where it is applicable, could provide a cheaper method of estimating long-term uplift rates (compared to thermochronology, cosmogenic nuclides, etc.). It is significant to the field of tectonic geomorphology in its focus on nonuniform uplift at the orogen scale (rather than an individual fault).

I had a few issues / questions that I would like the authors to address, none of which should impede the eventual publication of the manuscript.

1. In the introduction (lines 45 - 50), the authors suggest that their methods will enable us to estimate uplift gradients cheaply, such that the framework they propose could replace existing methods. Given the stringent rules of applicability (symmetrical lithology and precipitation across the divide), I think the authors oversell the utility /impact of their findings for natural systems. It is quite common, for example, for mountain ranges to induce an asymmetric pattern in precipitation, which would invalidate the methods discussed here. Indeed, this is evident in the acknowledgements where several people provided advice on selecting natural landscapes, suggesting that many natural landscapes would not serve as adequate examples. Furthermore, the position and mobility of the divide can only be used to directly calculate the uplift field if the uplift gradient is linear (line 193). I am not familiar enough with the literature on uplift gradients across mountain ranges to say for sure, but it did not seem obvious to me why the gradient should be linear (as opposed to nonlinear, or even a step function). I would like to see the authors discuss why they selected a linear gradient.

2. In the caption of figure 1, the authors note that it takes their models 300 Myr to attain steady state. This statement left me wondering about the typical timescale for orogenic steady state and how this timescale compares to the persistence of uplift gradients. A more explicit discussion of response timescales would be welcome.

3. Looking at figure 2, I'm not entirely convinced by the claim that the Taiwanese example has symmetrical precipitation and lithology across the divide. I'm particularly unconvinced by the precipitation map, which appears to show a much steeper gradient on the southeast side compared to the northwest side. The subfigure showing the geology of Taiwan (2e) is very busy and the geology map is somewhat obscured by the uplift measurements, GPS readings, and swath outlines. I think it would be sufficient to describe the swath procedure in the text or caption and leave it out of the figure.

4. I have several questions about Figure 1.

a. It appears that the numerical model results are systematically lower on the plot than the analytical solution. Is there a reason for this?

b. Between Figure 1 a-f it appears that the scale of drainage basins changes substantially, with drainage basin size appearing to scale with uplift gradient. Why is that? Does it affect the results, particularly the use of these models to fit the k and b parameters of the numerical model (Line 372 - 373)?

c. The results from Sicily don't appear to fit the model particularly well at the scale of the figure. Is this real or only appears this way because Taiwan has a much broader range of uplift gradients? What should we make of mismatches between natural systems and the model predictions?

d. How common is it to have uplift gradients (linear or otherwise) across natural mountain ranges?

5. I found Figure 3 confusing. My interpretation is that the transient states are cases where the erosional response to uplift is out of equilibrium with the uplift (i.e., moderate erosion and high uplift or high erosion and moderate uplift). The divide moves to change the steepness of the landscape, thereby putting the erosion and uplift back in equilibrium. What I don't understand is what the authors mean by "asymmetric uplift dominat[ing] divide mobility" (line 154) as opposed to the erosion contrast dominating divide mobility. I would suggest that the authors revise the text to simplify this conceptual framework; the idea of erosion rate and uplift rate as completely separate factors is confusing, since erosion rates are responding to uplift rates.

6. There are a few parts of the methods section that I would like to see clarified. On line 358, it's not clear what the "boundary between river channel and hillslope" refers to; I assume this is the upper extent of the river network, i.e., the part that is closest to the divide. It would be helpful if the authors say this explicitly. Additionally, my guess is that the assumption that "the elevation on

both sides are the same" (line 368) is part of the steady-state assumption, but would also like to see that laid out explicitly. Finally, more detail on the fit of k_2 and b from the numerical models would be helpful, particularly given the difference in drainage basin scale (see item #4 above). What was the spread of the best-fit values? Did they vary systematically with uplift gradient? And finally, two very minor notes:

1. The spatial orientation of the steady divide cases in Figure 4 is oriented in the opposite direction to the steady state numerical runs in Figure 1. It would be easier to compare if they were oriented the same way (with the high uplift on the top boundary).
2. There are several places in the manuscript that need light copy-editing, particularly the insertion of articles. I leave this to the copy editor and/or authors.

Reviewer #2 (Remarks to the Author):

This paper tackles an interesting and important problem: how does large scale mountain asymmetry reflect gradients in tectonic uplift rate and how can morphologic state of a mountain belt be used to infer information regarding the present or past uplift pattern. Positive aspects of the paper include the fact that it is using large-scale geomorphic data and rigorous modeling, both numerical and analytical. The field could use more work at this scale and I would be happy to see more papers with this approach.

That said, I don't think this paper is ready for publication. The authors have set up a difficult, perhaps impossible, problem by limiting their study to cases controlled exclusively by a horizontal gradient in vertical uplift rate. This requires finding control examples with no variations in rainfall or rock type, and no horizontal tectonic motion. I'm not sure such examples exist. Currently, there are problems in both of the selected examples as I will outline below.

General Comments:

(1) The analytical work and numerical modeling is good, and I appreciated seeing it. It is the strongest part of the paper. My one complaint is that some more context could have been given. There is considerable work done on this and similar problems, and although most of the other papers appear in the reference list, there are no comparisons made or explanations as to how this work stands apart as original. Probably the most important related paper is Goren et al., *ESPL*, 2014, whose supplement addresses the same problem with both analytical and numerical models. The current work uses a linear gradient in uplift, whereas Goren et al used constant values over each side of the range, but the physical principles are the same and the results similar. This could at least be explained to give some context. If advection is addressed in more detail (as I will suggest it should) work by Miller (e.g. Miller et al., *JGR* 2007) is relevant and similar in goals and scale.

(2) Advection of topography is the biggest problem here. Topographic asymmetry is much more sensitive to advection than it is to gradients in uplift (see Goren et al, 2014), so it is very important that this effect either be controlled (proven to be zero) or brought into the problem. The authors here have tried to argue that advection is not important to their examples in Sicily or Taiwan, but I don't think they succeeded. In Sicily, the authors did not show horizontal GPS data, but velocities relative to Europe are 5 to 10 mm/yr (Palano et al., 2012). This is not advection, it is rock velocity, but this raises another problem. Advective velocities are not easy to estimate. The authors show this in their Taiwan example by accident, in that they have taken the GPS velocities, but applied them completely backwards. The advective velocity is NW to SE, not SE to NW as they argue. What they show in Figure 2 are the GPS vectors with respect to Eurasia, but this is not the advection velocity. The velocity needed for advection of topography is the velocity of the rock with respect to the Earth's surface, not a trivial quantity to measure, since the Earth's surface does readily lend itself to measurement. Rocks at the surface can be measured, but the surface itself is an Eulerian feature, moving independent of its material (rock) at the surface today. For an orogenic wedge like Taiwan, topographic steady state is maintained by an accretionary flux from the NW into the mountain belt. Rock moves to the SE through the wedge coming to the surface

somewhere between the two mountain fronts which are near the west coast of Taiwan and the Longitudinal Valley in eastern Taiwan (not the east coast, but that is a different point). The first point on the east side of the Taiwan wedge that is not moving with respect to the Earth's surface is in the Longitudinal Valley, so this is the appropriate reference frame for advection. If you want to use GPS to measure advection, it needs to be shown in this reference frame. All points in the mountains are moving SE with respect to this point. If this is not clear, see figures in Willett et al., *AJS*, 2001; Willett et al., *Geology*, 2006, or Miller et al., *JGR*, 2007. This implies that the asymmetry in Taiwan could be entirely due to advection, not differential uplift and the authors just lost one example.

In Sicily, the tectonic setting is not so clear – this is a complex region with transtension in the Messinian strait and compression offshore to the north and perhaps also in southern Sicily. Advection would be defined as the rock velocity with respect to the erosional boundary, which could be taken as the coastline, so to establish no advection, it is necessary to show that the coasts are stationary with respect to the underlying rock. It could be true, but at the moment there is no demonstration of this point in the paper, so advection is left open.

This will be a difficult problem because in any tectonic setting with high uplift rates, needed to establish a steady state mountain belt, there will be horizontal motion and the advection will need to be controlled. I am not convinced that there are settings with high vertical rates and no horizontal motion, so the authors may need to reconsider their theoretical framework to include the advection and try to come up with a theory to account for both.

(3) Other Taiwan issues. The eastern limit of the Taiwan orogen is the Longitudinal Valley, not the east coast. The coastal range is a different tectonic entity with the Longitudinal valley separating the two. The basin is a persistent feature since the onset of the collision and serves as the geomorphic boundary.

Using the vertical GPS data for uplift may be a mistake. These data are nearly an order of magnitude larger than all the erosion rate data (See Fellin et al., *Global Planet. Change*, 2017). If Taiwan is steady state, these should be the same. The erosion rate data are much more internally consistent and consistent with long term exhumation rates. Unfortunately, they will not have the resolution to show any uplift gradient, if there is one.

(4) Other Sicily issues. The Quaternary and Pleistocene uplift patterns are very different. As are the average rates. Makes it difficult to justify steady state and suggests that there should be big along strike variations in the divide position, which are not there. This needs explanation and better justification for steady state assumption and for along strike variability

(5) I'm not sure the point about the 5 states of divide dynamics is a very valuable direction to take the work as a final stage. For one thing it assumes that the past climate (precipitation) and advection components have been constant, in order to make a conclusion regarding the change in uplift rates. This is a more difficult assumption than the current assumption that the modern advection and precipitation are known. Also, there is no methodology description for the divide dynamics assessment.

Sean Willett

**Response Letter**

We thank the reviewers and the editor for their insightful and constructive comments and
corrections, which helped us to greatly improve the manuscript. We address their concerns
point by point, and highlight implemented changes in the manuscript.

**Responses to Reviewer #1**

This paper presents numerical and analytical evidence that, all else equal, the position and
mobility of the main divide of a mountain belt should be directly related to the uplift field
across the range. The analysis presented is elegant and, in the places where it is applicable,
could provide a cheaper method of estimating long-term uplift rates (compared to
thermochronology, cosmogenic nuclides, etc.). It is significant to the field of tectonic
geomorphology in its focus on nonuniform uplift at the orogen scale (rather than an individual
fault).

I had a few issues / questions that I would like the authors to address, none of which should
impede the eventual publication of the manuscript.

**Response: We thank the reviewer for the positive comments.**

1. In the introduction (lines 45–50), the authors suggest that their methods will enable us to
estimate uplift gradients cheaply, such that the framework they propose could replace existing
methods. Given the stringent rules of applicability (symmetrical lithology and precipitation
across the divide), I think the authors oversell the utility /impact of their findings for natural
systems. It is quite common, for example, for mountain ranges to induce an asymmetric
pattern in precipitation, which would invalidate the methods discussed here. Indeed, this is
evident in the acknowledgements where several people provided advice on selecting natural
landscapes, suggesting that many natural landscapes would not serve as adequate examples.

**Response: In the previous manuscript, we mainly focused on the relationship between uplift**
**gradient and the location of the main drainage divide (MDD) of a mountain belt. We agree**
**with the reviewers and the editor that the application of the previous models is limited to the**
**cases with uniform precipitation, lithology, and no advection. In the revised version, we have**
**included advection (by both numerical simulations and the analytical solution) in our models,**
**which obviously broadens the applicability. We have also incorporated a few examples of**
**mountains with heterogenous precipitation and/or lithology, for which our conceptual**
**frameworks are still useful to constrain tectonic information. For example, Southern Taiwan**
**has uplift gradient, advection, and precipitation differences on both sides, all these forces**
**push the MDD towards the SE. Based on the quantitative constraints for all relevant**
**parameters that influence the position of the MDD, our models suggest that the current divide**
**location value is much lower than the model predication (Fig. S8), we attribute this disparity**
**to a recent increase in tectonic activity (please see Lines 170–208 of the revised manuscript**
**for detailed interpretations).**

**Importantly, we state that our models can provide first-order constrains on tectonic**
**information, providing a basis to design more targeted data collection campaigns in further**
**investigations (Lines 353–358 of the revised manuscript).**

2. Furthermore, the position and mobility of the divide can only be used to directly calculate
the uplift field if the uplift gradient is linear (line 193). I am not familiar enough with the
literature on uplift gradients across mountain ranges to say for sure, but it did not seem
obvious to me why the gradient should be linear (as opposed to nonlinear, or even a step
function). I would like to see the authors discuss why they selected a linear gradient.

**Response:** The reviewer raises a valuable point. Most mountain ranges do have uplift
gradients, because faults and other tectonic boundaries have finite lengths and stress-driven
gradients. Only those created by regional or isostatic uplift may have no uplift gradient. There
are two reasons why we selected a linear gradient: one is that the linear formulation needs
fewer constraints, and the other is to make our results more readily comparable with
previously published works, which have also used linear uplift gradients in their numerical
models (e.g. *Goren et al., 2014; Willett et al., 2014; Whipple et al., 2017*). In addition, we
also tested the sensitivity of the divide location in response to nonlinear uplift gradient in the
previous manuscript. The results show that if the uplift gradient is nonlinear, our models can
still be used to constrain the pattern of tectonics, rather than obtain a quantitative gradient
value (Lines 333–351 of the revised manuscript). To emphasize this, we moved the related
figure from the supplementary information into the main text as Fig. 7.

3. In the caption of figure 1, the authors note that it takes their models 300 Myr to attain
steady state. This statement left me wondering about the typical timescale for orogenic steady
state and how this timescale compares to the persistence of uplift gradients. A more explicit
discussion of response timescales would be welcome.

**Response:** Thanks for this valuable suggestion. Based on the mean elevations of the modelled
topographies, we calculated the time needed to reach topographic steady state. Generally, the
time decreases with increasing gradients of uplift and advection (Figs. S4 and S5), in
agreement with the results of a previous numerical study (*Willett et al., 2001*). Due to the
existence of advection, no feature within the landscape is steady, except for the MDD (*Willett
et al., 2001*). Thus, the mean elevations show fluctuations for models with advection, even at
an overall steady state (Fig. S4) (Lines 117–121 of the revised manuscript).

4. Looking at figure 2, I'm not entirely convinced by the claim that the Taiwanese example
has symmetrical precipitation and lithology across the divide. I'm particularly unconvinced
by the precipitation map, which appears to show a much steeper gradient on the southeast side
compared to the northwest side.

**Response:** We agree to the reviewer's comment and have provided more detailed information
about the precipitation rate and rock erodibility in the revised manuscript. Based on the
precipitation rate data shown in Fig. 3e, we calculated the average annual precipitation rates
for the two sides. The average rates for the NW and SE sides of the Southern Taiwan are 2528
82 mm/yr and 2231 mm/yr, respectively. The slightly higher precipitation rate on the NW side
tends to push the divide towards the SE. We also estimated the erodibility of Southern Taiwan
to be $5.8 \times 10^{-5} \text{ yr}^{-1}$ and $4.1 \times 10^{-5} \text{ yr}^{-1}$ for the NE and SE sides, respectively (see Methods).
These results were integrated to calculate the divide location of Southern Taiwan based on the
current geological settings (Lines 512–528 of the revised manuscript).

5. The subfigure showing the geology of Taiwan (2e) is very busy and the geology map is
somewhat obscured by the uplift measurements, GPS readings, and swath outlines. I think it
would be sufficient to describe the swath procedure in the text or caption and leave it out of
the figure.

Response: Since the swath procedures for both Southern Taiwan and NE Sicily have been
addressed in the Methods section, following the reviewer's suggestion, we removed the
swaths in Southern Taiwan and NE Sicily from this figure (Fig. 3 in the revised manuscript).

6. I have several questions about Figure 1.

a. It appears that the numerical model results are systematically lower on the plot than the
analytical solution. Is there a reason for this?

Response: It's true that the analytical solution for uplift gradient is systematically higher than
the predictions from numerical models. Similar deviation can also be found in *Goren et al.*,
2014. We have added some extra sentences to explain this as:

*'However, for the same uplift gradient, the analytical solution systematically predicts higher*
*values of d than the results of numerical models. This may due to the Hack's law (Hack, 1957)*
*exponent b that is used for the analytical solution ($b = 1.6$, see Methods) does not fit exactly*
*with the simulations (Goren et al., 2014). Instead, it varies with uplift gradients (Figs. S1 and*
*S2). The results from the analytical solution demonstrate that the divide location generally*
*increases with increasing b (Fig. S3).'* (Lines 70–75 of the revised manuscript).

b. Between Figure 1 a-f it appears that the scale of drainage basins changes substantially, with
drainage basin size appearing to scale with uplift gradient. Why is that? Does it affect the
results, particularly the use of these models to fit the k and b parameters of the numerical
model (Line 372 – 373)?

Response: The main drainage divide determines the mountain width on both sides of the
mountain belt. With the migration of the main divide to the upper side, the mountain width of
the upper side decreases, whilst the value of the lower side increases. The mountain width
limits the length and area of drainage basins. This is the reason why drainage basin areas scale
with uplift gradients.

In the previous manuscript, we just manually selected some representative drainage
basins to calculate k_2 and b , with average values of 0.04 and 1.4, respectively. According to
the reviewer's suggestion, we analysed all the drainage basins with drainage areas greater
than 5 km² for the 20 numerical models with uplift gradients (Figs. S1, S2), 11 numerical
models with advection gradients (Figs. S2, S10), and NW and SE sides of Southern Taiwan
(Figs. S7). For each topography, we acquired hundreds of basins, and the associated k_2 , b , and
R^2 values (please see Source Data file for the drainage area and river length data).

The average values of k_2 and b for the 20 numerical models with uplift gradients are 0.7
and 1.6, respectively (Fig. S1). The average values of k_2 and b for the 11 numerical models
with advection are 0.6 and 1.4, respectively (Fig. S10). There is no systematic correlation
between uplift gradient and Hack's parameters (Fig. S2), which is supported by the
observations from natural landscapes that exhibit the self-similarity of drainage basins
(*Sassolas-Serrayet et al. 2018, Nature Communications*). By contrast, for the 11 models with
advection, the value of b decreases with increasing advection gradients (Fig. S2), because

higher advection gradients lead to more elongated drainage basins (Fig. 1d–f) (Lines 417–424
of the revised manuscript).

c. The results from Sicily don't appear to fit the model particularly well at the scale of the
figure. Is this real or only appears this way because Taiwan has a much broader range of
uplift gradients? What should we make of mismatches between natural systems and the model
predictions?

Response: The natural examples used to compare with the relationship between uplift
gradient and divide location need to meet some requirements. Besides, natural landscapes are
driven by variable forces, e.g., rainfall varying over space and time, as the reviewer
previously mentioned. This variability is reflected as the noise in the topography. It is likely
the reason why numerical model results and natural systems are not perfectly matched.
According to the reviewer #2, we only kept the Pleistocene uplift data, integrating over a
more meaningful timescale for the problem we are addressing.

147 d. How common is it to have uplift gradients (linear or otherwise) across natural mountain
ranges?

Response: Most mountain ranges do have uplift gradients, because faults and other tectonic
boundaries have finite lengths and stress-driven gradients. Only those created by regional or
isostatic uplift may have no uplift gradient. Please see Lines 49–60 of the Response Letter for
detail.

7. I found Figure 3 confusing. My interpretation is that the transient states are cases where the
erosional response to uplift is out of equilibrium with the uplift (i.e., moderate erosion and
high uplift or high erosion and moderate uplift). The divide moves to change the steepness of
the landscape, thereby putting the erosion and uplift back in equilibrium. What I don't
understand is what the authors mean by "asymmetric uplift dominat[ing] divide mobility"
(line 154) as opposed to the erosion contrast dominating divide mobility. I would suggest that
the authors revise the text to simplify this conceptual framework; the idea of erosion rate and
uplift rate as completely separate factors is confusing, since erosion rates are responding to
uplift rates.

Response: We thank the reviewer's suggestion, and have added the following sentences to
make the point clear as:

*'Erosion rates are controlled by tectonic deformation, but tectonic movements can start or*
*increase instantaneously, while erosion rates take some time to adjust to those changes. This*
*leads to some temporary decoupling between tectonic forcing and erosion, and hence during*
*some transient periods, one of these factors can be the main factor controlling divide*
*mobility.'* (Lines 214–218 of the revised manuscript).

8. There are a few parts of the methods section that I would like to see clarified.

a. On line 358, it's not clear what the "boundary between river channel and hillslope" refers
to; I assume this is the upper extent of the river network, i.e., the part that is closest to the
divide. It would be helpful if the authors say this explicitly.

Response: Thanks for pointing this out. At some points, river channels start to form. These
points are termed as the boundary between hillslope and river channel. Because we did not
include hillslope in the analytical solution for uplift gradient, we have changed ‘the elevation
at the boundary between river channel and hillslope are the same’ into ‘the elevation at the
MDD are the same’ (Lines 404, 406, and 414 of the revised manuscript).

b. Additionally, my guess is that the assumption that “the elevation on both sides are the same”
(line 368) is part of the steady-state assumption, but would also like to see that laid out
explicitly.

Response: Yes, previous works (e.g., *Goren et al., 2014*) have also assumed that in steady
state, at hillslope-channel boundary, the elevation on both sides are the same. This sentence
has been revised as ‘At the MDD, the elevation on both sides are the same.’ (Line 414 of the
revised manuscript).

c. Finally, more detail on the fit of k_2 and b from the numerical models would be helpful,
particularly given the difference in drainage basin scale (see item #4 above). What was the
spread of the best-fit values? Did they vary systematically with uplift gradient?

Response: According to reviewer’s suggestion, more detail on the fit of k_2 and b from the
numerical models and Southern Taiwan has been added (Figs. S1, S2, S7, S10). Please see
Lines 118–133 of the Response Letter for detail.

And finally, two very minor notes:

11. The spatial orientation of the steady divide cases in Figure 4 is oriented in the opposite
direction to the steady state numerical runs in Figure 1. It would be easier to compare if they
were oriented the same way (with the high uplift on the top boundary).

Response: We agree to the reviewer’s suggestion that consistency of direction would be easier
to compare. At the same time, we also need to consider the arrangement of our figure, i.e.,
with enough space to type the words. Accordingly, we have revised Fig. 4a–d, whereas Fig.
4e remains as previously shown. We believe the revised figure is concise enough to be
understand (Fig. 5 in the revised manuscript).

12. There are several places in the manuscript that need light copy-editing, particularly the
insertion of articles. I leave this to the copy editor and/or authors.

Response: We have carefully checked the manuscript. Besides, the revised main text and the
Supplementary Information have been edited by *Nature Research Editing Service*, which has
improved the readability of our manuscript.

**Responses to Reviewer #2 (Sean Willett)**

This paper tackles an interesting and important problem: how does large scale mountain
asymmetry reflect gradients in tectonic uplift rate and how can morphologic state of a
mountain belt be used to infer information regarding the present or past uplift pattern. Positive
aspects of the paper include the fact that it is using large-scale geomorphic data and rigorous
modeling, both numerical and analytical. The field could use more work at this scale and I
would be happy to see more papers with this approach.

Response: Thanks for the positive comments.

That said, I don't think this paper is ready for publication. The authors have set up a difficult,
perhaps impossible, problem by limiting their study to cases controlled exclusively by a
horizontal gradient in vertical uplift rate. This requires finding control examples with no
variations in rainfall or rock type, and no horizontal tectonic motion. I'm not sure such
examples exist. Currently, there are problems in both of the selected examples as I will
outline below.

Response: Following the reviewer's suggestion, we have added 13 extra numerical
simulations with advection gradients (Figs. 1 and 2). Among them, 11 simulations for
mountains with a uniform uplift and variable advection gradients, 2 simulations for mountains
with both uplift and advection gradients. Furthermore, following the advection theory set up
by Goren et al. (2014), we also worked out the analytical solution for divide location with
respect to advection gradients. We provide the details in Lines 330–346 of the Response
Letter.

We have further discussed the natural landscapes of Southern Taiwan and NE Sicily,
which will be explained in more detail later (lines 349–403 of the Response Letter).

We have not only added the results of numerical simulations and analytical solution for
advection, but also improved our conceptual models (Figs. 4–6). We believe our revised
models would have broader applications, as explained in Lines 425–453 of the Response
Letter.

**General Comments:**

(1) The analytical work and numerical modeling is good, and I appreciated seeing it. It is the
strongest part of the paper. My one complaint is that some more context could have been
given. There is considerable work done on this and similar problems, and although most of
the other papers appear in the reference list, there are no comparisons made or explanations as
to how this work stands apart as original. Probably the most important related paper is Goren
et al., ESPL, 2014, whose supplement addresses the same problem with both analytical and
numerical models. The current work uses a linear gradient in uplift, whereas Goren et al used
constant values over each side of the range, but the physical principles are the same and the
results similar. This could at least be explained to give some context. If advection is addressed
in more detail (as I will suggest it should) work by Miller (e.g. Miller et al., JGR 2007) is
relevant and similar in goals and scale.

Response: This work was motivated by the results of numerical studies carried out by Goren
et al., 2014, Willett et al., 2014, and Whipple et al., 2017, demonstrating that the main divide
will migrate in response to asymmetric uplift. We agree on the necessity of comparisons with

the results of these works. Accordingly, we have added extra sentences addressing this issue
in the revised manuscript. We list them below in the order of lines in which they appear in the
main text:

- (a) *For a spatially uniform advection rate of 0.5 mm/yr, the MDD forms near the centre ($d =$
$49.7 \pm 3.5\%$) of mountain range (Fig. 1a), consistent with previous works (Willett et al.,
2001, 2014; Goren et al., 2014) (Lines 60–62 of the revised manuscript);*
- (b) *For a spatially uniform advection rate of 0.5 mm/yr, the MDD forms at $d = 55.6 \pm 6.0\%$
(Fig. 1g), in agreement with previous works (Willett et al., 2001; Miller et al., 2007;
Goren et al., 2014) (Lines 81–82 of the revised manuscript);*
- (c) *Generally, the time needed to reach steady state decreases with increasing gradients of
uplift and advection (Figs. S4 and S5), in agreement with the results of a previous
numerical study (Willett et al., 2001). Due to the existence of advection, no feature within
the landscape is steady, except for the MDD (Willett et al., 2001). Thus, the mean
elevation shows fluctuations for models with advection, even at an overall steady state
(Fig. S4). (Lines 117–121 of the revised manuscript);*
- (d) *The MDD migrates to the side with a higher uplift rate (Goren et al., 2014; Willett et al.,
2014; Whipple et al., 2017; He et al., 2019), lower precipitation and/or rock resistance
(Bonnet et al., 2009; Goren et al., 2014; Willett et al., 2014; Forte et al., 2015, 2018;
Whipple et al., 2017), and in the direction of advection (Willett et al., 1999, 2001, 2002;
Miller et al., 2007; Goren et al., 2014; Forte et al., 2015). (Lines 136–137 of the revised
manuscript);*
- (e) *For the orogenic wedge of Southern Taiwan, rock moves from the NW to the SE through
the mountain belt coming to the surface near the Longitudinal Valley (Fig. 3d,e) (Willett
et al., 2001) (Lines 173–174 of the revised manuscript);*
- (f) *The average advection gradient in Southern Taiwan is 0.35 mm/yr/km (see Methods),
which pushes the MDD towards the SE (Willett et al., 2001) (Lines 176–178 of the
revised manuscript);*
- (g) *We attribute this disparity to a recent increase in tectonic activity (rates of uplift and
advection), which is likely due to the southward propagation of the arc-continent
collision in Taiwan (Suppe, 1981; Willett et al., 2002, 2003; Fellin et al., 2017). Erosion
rates derived from low-temperature thermochronometry and cosmogenic nuclide over
millennial time scales (Fellin et al., 2017) are much lower than GPS-derived uplift rates
(Fig. 3d), supporting this interpretation (Lines 186–190 of the revised manuscript);*
- (h) *Similarly, in the Southern Alps of New Zealand, the uplift gradients and advection push
the MDD to the NW, despite a significant precipitation rate differences on the two sides
(the precipitation rates on the NW and SE sides of the range are ~ 12 m/yr and 1 m/yr,
respectively) that would push the MDD to the SE (Goren et al., 2014) (Lines 205–208 of
the revised manuscript).*
- (i) *The advection theory set up is the same as that of Goren et al. (2014) (Lines 429–461 of
the revised manuscript).*

(2) Advection of topography is the biggest problem here. Topographic asymmetry is much
more sensitive to advection than it is to gradients in uplift (see Goren et al, 2014), so it is very
important that this effect either be controlled (proven to be zero) or brought into the problem.

The authors here have tried to argue that advection is not important to their examples in Sicily
or Taiwan, but I don't think they succeeded. In Sicily, the authors did not show horizontal
GPS data, but velocities relative to Europe are 5 to 10 mm/yr (Palano et al., 2012). This is not
advection, it is rock velocity, but this raises another problem. Advective velocities are not
easy to estimate. The authors show this in their Taiwan example by accident, in that they have
taken the GPS velocities, but applied them completely backwards. The advective velocity is
NW to SE, not SE to NW as they argue. What they show in Figure 2 are the GPS vectors with
respect to Eurasia, but this is not the advection velocity. The velocity needed for advection of
topography is the velocity of the rock with respect to the Earth's surface, not a trivial quantity
to measure, since the Earth's surface does readily lend itself to measurement. Rocks at the
surface can be measured, but the surface itself is an Eulerian feature, moving independent of
its material (rock) at the surface today. For an orogenic wedge like Taiwan, topographic
steady state is maintained by an accretionary flux from the NW into the mountain belt. Rock
moves to the SE through the wedge coming to the surface somewhere between the two
mountain fronts which are near the west coast of Taiwan and the Longitudinal Valley in
eastern Taiwan (not the east coast, but that is a different point). The first point on the east side
of the Taiwan wedge that is not moving with respect to the Earth's surface is in the
Longitudinal Valley, so this is the appropriate reference frame for advection. If you want to
use GPS to measure advection, it needs to be shown in this reference frame. All points in the
mountains are moving SE with respect to this point. If this is not clear, see figures in Willett
et al., *AJS*, 2001; Willett et al., *Geology*, 2006, or Miller et al, *JGR*, 2007. This implies that
the asymmetry in Taiwan could be entirely due to advection, not differential uplift and the
authors just lost one example.

**Response:** We agree that advection is necessary to be included in our models, as mountain
asymmetry is very sensitive to advection (e.g., Willett et al., 2001; Miller et al., 2007; Goren
et al., 2014). According to reviewer's advice, we have added 11 extra numerical models with
a uniform uplift but varying in advection gradients (Fig. 1). Moreover, we worked out the
analytical solution for divide location with respect to advection gradient (Lines 429–461 of
the revised manuscript), which fits well with the numerical results (Fig. 1g). The results show
that the advection will push the divide in the direction of advection, consistent with previous
works (e.g. Willett et al., 1999, 2001, 2002; Miller et al., 2007; Goren et al., 2014; Forte et al.,
2015). Furthermore, our results also demonstrate that the asymmetry of a mountain belt
increases with advection gradients, consistent with the results of previous numerical
simulations (Willett et al., 2001). Another two numerical models consider mountain belts with
uplift gradient and advection simultaneously (Fig. 2). When they push the MDD in the same
direction, the mountain becomes more asymmetric (Fig. 2a-c). By contrast, when they push
the divide in different directions, their effects partially cancel out. In this case, the steady-state
divide location is dominated by the driver with a higher contribution to divide migration, and
the mountain belt is less asymmetric (Fig. 2a,b,d). Please see Lines 106–115 of the revised
manuscript for details.

Accordingly, we have improved our conceptual models (Figs. 4–6), which we believe
have wider applications (Lines 306–332 of the revised manuscript).

Furthermore, we have discussed about the natural examples of NE Sicily and Southern
Taiwan in more detail (Lines 135–208 of the revised manuscript). The mistake on using GPS

direction as the advection pointed out by the reviewer has been corrected. Following the
reviewer's suggestion, we have also rechecked the tectonic background of the NE Sicily and
Southern Taiwan.

**For NE Sicily**, following the reviewer's advice, we added GPS horizontal velocities in
Fig. 3a based on the data of Palano et al., (2012), and tried to explain the advection in this
area as:

*'The uplift of NE Sicily is controlled by the activity of the Messina-Taormina Fault (MTF, Fig.*
*3a). NE Sicily, on the footwall of the MTF, is relatively stable compared with the hanging*
*wall. Hence, we infer that the advection is insignificant here, as supported by the relatively*
*uniform GPS velocities, moving towards N-NE at a rate of approximately 4.4 ± 1.3 mm/yr in a*
*fixed central Europe frame (Palano et al., 2012)'* (Lines 149–153 of the revised manuscript).

**For Southern Taiwan**, we modified the text as:

*'We acquired all the parameters that can influence divide mobility for Southern Taiwan (see*
*Methods). The slightly higher precipitation rate on the NW side pushes the MDD towards the*
*SE. For an orogenic wedge as Southern Taiwan, rock moves from the NW to the SE through*
*the mountain belt coming to the surface near the Longitudinal Valley (Fig. 3d,e) (Willett et al.,*
*2001). Thus, we chose the Longitudinal Valley as the reference frame for advection. Other*
*points in the mountains are moving towards the SE with respect to the valley (Fig. 3d). The*
*average advection gradient in Southern Taiwan is 0.35 mm/yr/km (see Methods), which*
*pushes the MDD towards the SE (Willett et al., 2001). The uplift gradient (average value of*
*0.14 mm/yr/km) also pushes the MDD towards the SE (Figs. 1-3). The uplift gradient,*
*advection, and precipitation all push the MDD towards the SE. Although the d - λ data points*
*generally follow the trend of the modelled results and analytical solution, these points are still*
*lower than the model prediction that includes uplift gradients only (Fig. 1g). Furthermore,*
*based on the current average advection and uplift gradients, precipitation, mountain width,*
*Hack's parameters (Fig. S7), hillslope length, and erodibility constrained from channel*
*steepness, we numerically integrated the gradient function in the stream power model to*
*estimate the steady-state location at $d = 83.3\%$ (Fig. S8), which is much higher than the*
*current value of 64.7%. We attribute this disparity to a recent increase in tectonic activity*
*(rates of uplift and advection), which is likely due to the southward propagation of the*
*arc-continent collision in Taiwan (Suppe, 1981; Willett et al., 2002, 2003; Fellin et al., 2017).*
*Erosion rates derived from low-temperature thermochronometry and cosmogenic nuclides*
*over millennial time scales (Fellin et al., 2017) are much lower than GPS-derived uplift rates*
*(Fig. 3d), supporting this interpretation. This implies that although the topography has*
*reached steady state over shorter timescales (Fig. S6), the mountain belt is in a transient state*
*over longer timescales. We can expect that under persistent conditions, the MDD of Southern*
*Taiwan will migrate southeastwards to reach a divide location at $d \approx 83.3\%$.*

*All the relevant parameters for Southern Taiwan allow the assessment of the relative*
*importance of various controls on the position of the MDD (Fig. S8). If tectonic deformation*
*does not contribute to divide position, i.e., uniform uplift and no advection, the MDD forms*
*near the centre of the mountain belt with $d = 50.9\%$. Including advection only, the MDD*
*forms at $d = 70.9\%$, which is higher than the observed $d = 64.7\%$. Including uplift gradient*
*only, the MDD forms at $d = 76.9\%$. Including both advection and uplift gradients, the MDD*
*forms at $d = 83.3\%$. This value only slightly decreases to $d = 82.2\%$ for equal precipitation*

*on both sides of the range. Imposing a huge precipitation difference with values of 2 m/yr and*
*12 m/yr on the NW and SE sides, respectively, the mountain belt is symmetric with $d = 49.6\%$.*
*Thus, the tectonic deformation dominantly determines the divide location, while precipitation*
*is the secondary control. Similarly, in the Southern Alps of New Zealand, the uplift gradient*
*and advection push the MDD to the NW, despite significant precipitation rate differences on*
*the two sides (the precipitation rates on the NW and SE sides of the range are ~ 12 m/yr and*
*1 m/yr, respectively) that would push the MDD to the SE (Goren et al., 2014).’ (Lines 170–*
*208 of the revised manuscript).*

(3) In Sicily, the tectonic setting is not so clear – this is a complex region with transtension in
the Messinian strait and compression offshore to the north and perhaps also in southern Sicily.
Advection would be defined as the rock velocity with respect to the erosional boundary,
which could be taken as the coastline, so to establish no advection, it is necessary to show that
the coasts are stationary with respect to the underlying rock. It could be true, but at the
moment there is no demonstration of this point in the paper, so advection is left open.
This will be a difficult problem because in any tectonic setting with high uplift rates, needed
to establish a steady state mountain belt, there will be horizontal motion and the advection
will need to be controlled. I am not convinced that there are settings with high vertical rates
and no horizontal motion, so the authors may need to reconsider their theoretical framework
to include the advection and try to come up with a theory to account for both.

*Response: We agree with the reviewer that more introduction about the tectonic setting of NE*
*Sicily is needed. Thus, we added GPS horizontal velocities in NE Sicily, and further*
*discussed the advection here. Please see Lines 354–362 of the Response Letter for detail.*

*In the previous manuscript, the application of our models is limited to the cases with*
*uniform precipitation, lithology, and no advection. As the reviewer has pointed out that the*
*uplift is generally accompanied by advection. We added 13 numerical simulations and an*
*analytical solution to show how the MDD responds to advection (Figs. 1 and 2). Please see*
*Lines 330–346 of the Response Letter for detail. Accordingly, we have also included*
*advection in the concept models (Figs. 4–6).*

*Based on these modifications, we re-evaluated the application of our models as:*
*‘These ten cases (Figs. 5 and 6) cover all possibilities and can be widely used to constrain*
*tectonic information for all mountain belts that fulfil our assumptions, just based on the*
*present location and mobility of the MDD. For uniform precipitation and lithology, there are*
*four main situations that can use our models to constrain tectonic information. First, for*
*mountain belts with uplift gradients and negligible advection, constraints on uplift can be*
*obtained according to conceptual considerations in Fig. 5. Second, similarly, for mountain*
*belts with advection and uniform uplift, constraints on advection can be obtained according*
*to the conceptual considerations in Fig. 6. Third, if both uplift gradient and advection push*
*the MDD towards the same direction (Fig. 2c), the divide location can be compared with the*
*relationship for uplift and/or advection gradients (Fig. 1g). If the measured d values are*
*below the theoretical prediction, then the tectonic activity is constrained to have recently*
*increased (i.e., the case study of Southern Taiwan). Fourth, when uplift gradient and*
*advection push the MDD in opposing directions (Fig. 2d), the models provide lower bounds*
*for uplift or advection gradients based on the steady-state divide location (Fig. 1g).*

*Our models can provide constraints on tectonic information for many natural landscapes,*
*including those with heterogeneous precipitation and/or lithology. Southern Taiwan is an*
*example. If the MDD of a symmetric mountain belt is moving towards the side with higher*
*precipitation and/or weaker lithology (i.e., greater erosion rates), our conceptual models can*
*indicate the presence of an uplift and/or advection gradient pushing the MDD to the other*
*side and maintaining the symmetry. For the NE Sicily example, the present MDD is stable*
*(Fig. S6) and not in the centre of the mountain belts (Fig. 3). Thus, its present uplift is higher*
*on the steeper side (Fig. 5c), and the present uplift gradient can be obtained from Fig. 1g.*
*For the Wula Shan horst in northern China, which also has negligible advection, the uplift*
*rate is higher at the southern edge and decreases to the northern edge (He et al., 2019), and*
*the present divide location d is 60.8% (Fig. S9). Thus, its present uplift gradient can be*
*estimated at 0.008 mm/yr/km (equation 10). With a mountain width of 17.5 km (Fig. S9), the*
*difference in uplift rate between the southern and northern edge of the Wula Shan horst is*
*0.14 mm/yr.’ (Lines 306–332 of the revised manuscript).*

(4) Other Taiwan issues. The eastern limit of the Taiwan orogen is the Longitudinal Valley,
not the east coast. The coastal range is a different tectonic entity with the Longitudinal valley
separating the two. The basin is a persistent feature since the onset of the collision and serves
as the geomorphic boundary.

*Response: The Longitudinal valley is the eastern boundary of the Taiwan Orogen. But, in our*
*numerical simulations and the analytical solution for the models with uplift gradients, the four*
*boundaries of the mountain belt have the same elevation, serving as base level. If we use the*
*Longitudinal valley as boundary, the problem of differential base levels may arise. By*
*contrast, for advection, following the reviewer’s advice, we chose the Longitudinal Valley as*
*the reference frame for advection. In this regard, other points in the mountains are moving*
*towards the SE with respect to the valley (Fig. 3d).*

(5) Using the vertical GPS data for uplift may be a mistake. These data are nearly an order of
magnitude larger than all the erosion rate data (See Fellin et al., Global Planet. Change, 2017).
If Taiwan is steady state, these should be the same. The erosion rate data are much more
internally consistent and consistent with long term exhumation rates. Unfortunately, they will
not have the resolution to show any uplift gradient, if there is one.

*Response: As the reviewer pointed out, based on long-term exhumation rates of*
*low-temperature thermochronometry, there is no such high-resolution data to show uplift*
*gradient in the study area. Due to the existence of advection and precipitation difference*
*between the two sides, the uplift gradient is only used to provide cursory information. Besides,*
*the fact that the erosion rates derived from low-temperature thermochronometry and*
*cosmogenic nuclide that integrated over a longer time scale (Fellin et al., 2017) are much*
*lower than the GPS-derived uplift rates yields insight into the tectonic activity that may have*
*increased in recent geological time (please see Lines 188–190 of the revised manuscript for*
*detail).*

(6) Other Sicily issues. The Quaternary and Pleistocene uplift patterns are very different. As
are the average rates. Makes it difficult to justify steady state and suggests that there should

be big along strike variations in the divide position, which are not there. This needs
explanation and better justification for steady state assumption and for along strike variability.
Response: Following the reviewer's advice, we only kept the Pleistocene ones (Figs. 1 and 3)
that integrate over a more meaningful timescale for the problem we are addressing. The
overall topography of NE Sicily is in steady state, which is supported by the topographic
metrics near the divide (Fig. S6). Indeed, both uplift rates and divide location vary along
strike. Thus, we divided the study area into 10 main swaths, each with a width of 5 km to
reduce the effects of lateral variations (please see Lines 471–477 of the revised manuscript for
detail).

(7) I'm not sure the point about the 5 states of divide dynamics is a very valuable direction to
take the work as a final stage. For one thing it assumes that the past climate (precipitation)
and advection components have been constant, in order to make a conclusion regarding the
change in uplift rates. This is a more difficult assumption than the current assumption that the
modern advection and precipitation are known. Also, there is no methodology description for
the divide dynamics assessment.

Response: We agree to the reviewer's suggestion that including advection would significantly
broaden the applications of our models. For example, our revised models are useful even in
areas with uplift gradient, advection, and precipitation differences (such as Southern Taiwan).
Additionally, if the MDD of a symmetric mountain belt is moving towards the side with
higher precipitation and/or weaker lithology (i.e., greater erosion rates), our conceptual
models can indicate the presence of an uplift and/or advection gradient pushing the MDD to
the other side and maintaining the symmetry (please see Lines 311–338 of the revised
manuscript for more explanations about the application of our models). Besides, as
demonstrated in Southern Taiwan and Southern Alps of New Zealand (Goren *et al.*, 2014),
for tectonically active mountain belts, the position of the main drainage divide is dominantly
controlled by uplift gradient and advection, and that differential climatic conditions are of
minor importance (Lines 195–208 of the revised manuscript).

In the revised Fig. 4, we added divide migration in response to advection. Accordingly,
we revised the methodology description for the divide dynamics assessment as:

*'The tight relationship between d and λ , and d and γ (Fig. 1g), together with the conceptual*
*framework (Fig. 4), suggests that we can constrain tectonic information from the location and*
*mobility of the MDD. Both of these parameters can easily be obtained from the present*
*topography (Forte *et al.*, 2018).'* (Lines 252–255 of the revised manuscript).

REVIEWER COMMENTS

Reviewer #1 (Remarks to the Author):

He and coauthors have made thorough changes in response to my comments and to the comments of Dr. Willett – thank you! I particularly appreciate their examination of the sensitivity of the k and b parameters. The clarity of the manuscript and figures have been improved considerably. Most of all, I was impressed by their incorporation of advection in their model results and conceptual framework, which has increased the applicability of their results to natural landscapes. After reading the revised manuscript, I see no further need for revision and recommend that the manuscript be accepted for publication.

Reviewer #2 (Remarks to the Author):

Summary:

In this revised version of the paper, the authors have extended their analysis of mountain range divide asymmetry to include horizontal advection in their analytical treatment of the problem, conducting a large number of numerical experiments to demonstrate and test the analytical work and again considered implications and examples. In principle, this improves the applicability of the model by including more common tectonic kinematics, but I am concerned that the additional complexity of the problem has introduced more problems, rather than solving the old ones.

First, I think there are some problems with the way in which advection has been included in the paper, but first, I want to step back and look at the point of the paper. The analytical work in the paper is nicely put together and presented, but it is largely derivative of work by Goren et al. (2013), Miller et al., (2006), and probably a few others. There is value added by bringing this together with a common framing and goals of determining steady divide position, but it is only marginally original. The comparison to numerical models is interesting, but I'm not sure any longer of the purpose. Is it to confirm the analytical solutions or the numerical models? Analytical models are generally used to test numerical models, not the other way around, but I could see how the numerical model could be a test of the analytical assumptions regarding drainage basin geometry, i.e. reproducing scaling such as Hack's Law. The problem here is once advection is added, the numerical models do a terrible job at represented nature. This is a dirty little secret of landscape models, but in fact, they do terribly at tectonic problems with horizontal motion. This was one of the points of the Goren et al, 2013 paper. By using a regular rectilinear spatial grid and simple numerical methods (Landlab), this paper is not advancing this problem and is probably not really confirming either the analytical or numerical models. This means that the paper's main value is in the natural examples and showing the utility of the analysis in the real world. As I discuss below, there are problems in characterizing advection. It is good that this version of the paper has attempted to bring advection into the problem, but this is a tough problem and I don't think the authors have quite gotten it right.

Summary would be: advection needs to be fixed, point of the numerical models needs to be clarified and the long narrow basins explained or numerical models should be removed; natural examples need to be reanalyzed, bringing in data to constrain advection properly.

General Points:

(1) the advection is still not implemented correctly. The analytical treatment of the problem is correct, though largely based on the previous work by Goren et al., (2013) However, the authors have taken an overly complex parameterization. The simplest form of advection would be to add a constant horizontal velocity to the landscape, with boundary conditions of fixed position as well as elevation. However, the authors have chosen to implement, not a constant, but a linearly variable velocity (Eq. 13). A linearly variable velocity corresponds to a constant rate of shortening, i.e. constant strain rate, which is fine, but it needlessly complicates the problem by adding two parameters instead of one. The two parameters are now the velocity at one boundary (V_p in 13)

and the rate of change of that velocity with distance, or what they call the advection gradient. This raises the first problem: semantics. "Advection gradient" and "advection rate" not really appropriate terms. Advection is the process by which a physical quantity (heat, mass, topography) is carried by a moving material. The parameter quantifying this process is "velocity". Nearly everywhere in the paper, the term "advection" should be replaced by "velocity". What is imposed in the model is a "velocity gradient". This is just semantics, but should be corrected. More important is the way in which the gradient has been emphasized throughout the paper, when it is the fixed-point velocity that is the first order parameter. The paper largely addresses the "advection gradient", using this as the characteristic parameter, with the implication that it is the gradient in velocity that causes divide migration, much as the gradient in uplift does. This is not the case – any horizontal velocity, even with no gradient will drive divide migration. The first order parameter is going to be some metric, such as the mean horizontal velocity. It is not wrong to use a velocity that varies linearly with distance, but the problem then has two parameters and both need to be investigated. In most of the figures, e.g. 1, 2, S2, S5, only the variation with gradient is shown, not the variation with the fixed point. In particular, the important figures, 1, and 2 are not showing the full range of variation implicit to the advection, by holding V_p fixed. This also needs to be fixed.

I'm not sure why the authors chose to use a shortening rate instead of a constant velocity. It is more general than the constant velocity case, but more difficult to characterize. By adding another parameter, they need to show, and constrain, another dimension in parameter space, so simple summarized like Figure 1 will no longer be possible. This also means that any real applications will need further independent constraints.

The second problem with advection is that the natural examples are still not calculating the advection velocity correctly. The Taiwan example is more or less correct in that they have the correct velocity frame of reference now. However, the Sicily example still lacks a frame of reference, which will determine the mean value of the advection velocity. Currently the authors show that the velocity field has no gradient, but they take this to imply that advection is therefore negligible. Noting point above, this is not valid. It is the mean velocity with respect to the Earth's surface (assumed to be steady), not the gradient that is important. To estimate the advection velocity in Sicily, as in Taiwan, it is necessary to find the velocity of the rock with respect to the surface. If the surface is in steady state, that means the boundaries at the coast are the appropriate reference frame. The authors need to give the velocity of the coast with respect to the rock. At the moment there is no attempt to do this, but without this number, the advection velocity is unknown. I would guess, given the proximity of the coast to the MTF, that it is nearly fixed with respect to the hanging wall of that fault, and therefore the advection velocity is close to the slip rate on the fault, but the authors need to do some analysis of this.

There is a third problem with the treatment of advection and that is the way the divide mobility statistics have been applied (Fig. S6). These statistics were developed for regions undergoing uplift only, without advection and are either not valid or must be interpreted differently for settings with advection. For a setting with steady state topography and advection, the divide MUST move at the advection velocity, but in the other direction. Right? Otherwise it is not steady under the advection velocity. Interpretation of the geomorphic indicators must reflect this, although it depends somewhat on whether they are local (relief, slope at divide) or global (χ). If indicators really all show divides are stable (they don't look very stable to me), and there is steady state, there is no advection. Incidentally, this is also somewhat true with uplift gradients. Particularly χ values, but also steepness index needs to be corrected for uplift gradient.

(2) The Hacks Law problem described starting at line 70 is pretty well known in the landscape modeling community, at least among those of us who have worked on advection. The models in figure 1e, and f are not realistic landscapes. The advection has led to the creation of nearly perfectly uniform width, unbranching rivers, so the Hack Exponent on the Pro-wedge rivers is not correct and as noted is a function of how much advection is present. This raises the question as to what is the point of these numerical models. If they are to confirm the analytical model assumption that Hack's Law holds, they need first to reproduce Hack's Law. Otherwise I don't see the purpose of these models any longer.

Detailed comments by line number:

(1) Line 30-45: I'm not convinced that cost effectiveness is a very good selling point for this analysis. Divide position depends on uplift gradient, precip gradient, advection velocity and lithology, so the problem is multidimensional already and most of these methods need to be applied to constrain the problem anyway. Why not sell it on just learning something more – applying another constraint for internal self-consistency? Something fundamental. It would feel less contrived.

(2) Line 70. There is something interesting going on with the hack coefficients and exponents, but what? Why are they different? It would help to break out the Pro-wedge and retro-wedge rivers separately to see more.

(3) Line 117. I think other studies, including ref 5, show the opposite. Adding advection increases time to SS.

(4) Line 149. See above. This is not advection velocity. This is shortening rate or rock velocity with respect to Europe. This needs to be put in the reference frame of the land surface, as was done with Taiwan. To do this, you must find a point where the land surface is not moving with respect to the Eulerian surface that defines the land surface. Advection is complicated.

(5) Line 177. Just one example of problem throughout, but it is the advection velocity that pushed the divide, not the advection gradient.

(6) Line 188. Erosion rates from ^{10}Be are averaging over less than 10ka. For the GPS uplift data to be consistent means that tectonic uplift rates have increased by 2 to 10 times over a few thousand years over all of south Taiwan. This is not realistic. I think it is likely that the GPS are either wrong (vertical rates are hard to measure and signals always come down as more data are collected). Or they are dominated by a large interseismic elastic accumulation.

(7) Line 190. Topography cannot be steady over a short timescale and unsteady over a long timescale. This doesn't make sense. In addition the numerical models showed that steady state requires tens to hundreds of Ma to reach steady state, this hypothesized transient is occurring over a few thousand years. Cannot possibly be in steady state with the divide position reflected the GPS.

(8) Figures 4,5,6, : this is a strong and complicated emphasis on transients, when it is not clear how this will be constrained. With the multiple dimensionality of the problem, it is clear why it has become complicated, but not clear how this will be resolved. Much data, that generally does not exist would be needed and given that uplift, advection and rainfall are independent, it is not clear that real life cases will ever fall into these categories. Furthermore, there is no known way of establishing divide stability in the cases with advection. As discussed above, published divide metrics are only valid for cases without advection. I would recommend that this entire discussion be shortened and simplified.

(9) line 430: Willett et al., Nature, 2018 gives the advection solutions for general n , m , b , but no uplift.

Sean Willett

Response Letter

We thank the reviewers and the editor for their further insightful and constructive comments and suggestions. We address these concerns point by point, and highlight implemented changes in the manuscript.

Responses to Reviewer #1

He and coauthors have made thorough changes in response to my comments and to the comments of Dr. Willett – thank you! I particularly appreciate their examination of the sensitivity of the k and b parameters. The clarity of the manuscript and figures have been improved considerably. Most of all, I was impressed by their incorporation of advection in their model results and conceptual framework, which has increased the applicability of their results to natural landscapes. After reading the revised manuscript, I see no further need for revision and recommend that the manuscript be accepted for publication.

Response: We thank Reviewer #1 for positive comments and the recommendation of publication after the second round review of our manuscript.

Responses to Reviewer #2 (Sean Willett)

In this revised version of the paper, the authors have extended their analysis of mountain range divide asymmetry to include horizontal advection in their analytical treatment of the problem, conducting a large number of numerical experiments to demonstrate and test the analytical work and again considered implications and examples. In principle, this improves the applicability of the model by including more common tectonic kinematics, but I am concerned that the additional complexity of the problem has introduced more problems, rather than solving the old ones.

Response: Thank you for the insightful comments and constructive suggestions that greatly helped us to improve the clarity of the manuscript. Our response and corrections have been made point by point as follows.

General Points:

1. First, I think there are some problems with the way in which advection has been included in the paper, but first, I want to step back and look at the point of the paper. The analytical work in the paper is nicely put together and presented, but it is largely derivative of work by Goren et al. (2013), Miller et al., (2006), and probably a few others. There is value added by bringing this together with a common framing and goals of determining steady divide position, but it is only marginally original.

The comparison to numerical models is interesting, but I'm not sure any longer of the purpose.
Is it to confirm the analytical solutions or the numerical models? Analytical models are
generally used to test numerical models, not the other way around, but I could see how the
numerical model could be a test of the analytical assumptions regarding drainage basin
geometry, i.e. reproducing scaling such as Hack's Law. The problem here is once advection is
added, the numerical models do a terrible job at represented nature. This is a dirty little secret
of landscape models, but in fact, they do terribly at tectonic problems with horizontal motion.
This was one of the points of the Goren et al, 2013 paper. By using a regular rectilinear
spatial grid and simple numerical methods (Landlab), this paper is not advancing this problem
and is probably not really confirming either the analytical or numerical models. This means
that the paper's main value is in the natural examples and showing the utility of the analysis
in the real world. As I discuss below, there are problems in characterizing advection. It is
good that this version of the paper has attempted to bring advection into the problem, but this
is a tough problem and I don't think the authors have quite gotten it right.

**Summary would be:** advection needs to be fixed, point of the numerical models needs to be
clarified and the long narrow basins explained or numerical models should be removed;
natural examples need to be reanalyzed, bringing in data to constrain advection properly.

**Response:** Indeed, the idea that the main drainage divide (MDD) will migrate in response to
asymmetric uplift came from the numerical results of *Goren et al., 2014; Willett et al., 2014,*
*and Whipple et al., 2017*, and the related analytical work started by *Goren et al., 2014*.

Motivated by these excellent works, we aim to build a deeper linkage between tectonic
deformation and the location and mobility of the MDD from a more quantitative perspective.

The main contributions of this work are summarized as follows:

(1) Through stream power river incision model-based landscape evolution model and
analytical solution, we quantified the relationship between uplift gradient and the steady-state
location of the MDD;

(2) Using analytical solution, we quantified the relationship between advection velocity and
the steady-state location of the MDD;

(3) Using analytical solution, we quantified the relationship between the gradient of advection
velocity and the steady-state location of the MDD;

(4) Through landscape evolution model, we studied how the MDD of a mountain belt
responds to uplift gradient and advection, simultaneously. There are two situations: one is that
uplift gradient and advection push the MDD in the same direction (such as Southern Taiwan),
and the other is that they push the MDD in the opposite directions, like Northeastern Sicily;

(5) We built a conceptual framework exploring the interaction and competition between
tectonic uplift/advection and erosion in influencing the migration of the MDD;

(6) Based on the obtained quantitative relationships and conceptual framework, we built a
 model to constrain first-order tectonic information of a mountain belt based on the current
 location and mobility of the MDD;
 (7) We applied our model to Wula Shan horst, Northeastern Sicily, and Southern Taiwan to
 constrain their present or previous tectonic information.
 (8) We wrote a MATLAB script (divide_location.m) to generate a divide location under
 different settings (see Code availability), which can be used for other mountain belts. The
 results of the script and our analytical solutions match well (please see Fig. S13 below).

 Previously, we aimed to compare the 11 numerical simulations with constant shortening rates
 to the analytical solution. However, with higher advection velocity gradients in Fig. 3e,f of
 the previous manuscript, the drainage basins on the pro-wedge become long and narrow. The
 shapes of these drainage basins can hardly be observed in natural landscapes. As the reviewer
 has pointed out, they are not realistic landscapes. Another problem is that these simulations
 only considered advection velocity gradient. But, the average advection velocity is more
 important, which is determined by both the velocity gradient and the velocity at one boundary.
 Therefore, we have removed the results of the 11 numerical simulations in the main text,
 Supplementary Information, and Source Data file.

 Instead, through the analytical solutions, we not only quantified the relationship between the
 spatially uniform advection velocity and the steady-state divide location, but also quantified
 the relationship between the constant shortening rate (i.e., linear advection velocity gradient)
 and the steady-state divide location (please see the response to the second comment for
 detail).

 As for Northeastern Sicily, we appreciate and followed the reviewer's suggestions on how to
 choose the reference frame for advection. Accordingly, we chose two reference frames for
 advection and obtained velocities of 0.46 and 0.77 mm/yr, respectively (Fig. 4a). We also
 calculated the Hack's parameters (Fig. S12), average precipitation rate, hillslope length,

channel steepness, and erodibility for the Northeastern Sicily. Based on these data, we
re-analysed the Northeastern Sicily case (please see Lines 190–236 of the Response Letter for
detail).

2. The advection is still not implemented correctly. The analytical treatment of the problem is
correct, though largely based on the previous work by Goren et al., 2013. However, the
authors have taken a overly complex parameterization. The simplest form of advection would
be to add a constant horizontal velocity to the landscape, with boundary conditions of fixed
position as well as elevation. However, the authors have chosen to implement, not a constant,
but a linearly variable velocity (Eq. 13). A linearly variable velocity corresponds to a constant
rate of shortening, i.e constant strain rate, which is fine, but it needlessly complicates the
problem by adding two parameters instead of one. The two parameters are now the velocity at
one boundary (V_p in 13) and the rate of change of that velocity with distance, or what they
call the advection gradient. This raises the first problem: semantics. “Advection gradient” and
“advection rate” not really appropriate terms. Advection is the process by which a physical
quantity (heat, mass, topography) is carried by a moving material. The parameter quantifying
this process is “velocity”. Nearly everywhere in the paper, the term “advection” should be
replaced by “velocity”. What is imposed in the model is a “velocity gradient”. This is just
semantics, but should be corrected. More important is the way in which the gradient has been
emphasized throughout the paper, when it is the fixed-point velocity that is the first order
parameter. The paper largely addresses the “advection gradient”, using this as the
characteristic parameter, with the implication that it is the gradient in velocity that causes
divide migration, much as the gradient in uplift does. This is not the case – any horizontal
velocity, even with no gradient will drive divide migration. The first order parameter is going
to be some metric, such as the mean horizontal velocity. It is not wrong to use a velocity that
varies linearly with distance, but the problem then has two parameters and both need to be
investigated. In most of the figures, e.g. 1, 2, S2, S5, only the variation with gradient is shown,
not the variation with the fixed point. In particular, the important figures, 1, and 2 are not
showing the full range of variation implicit to the advection, by holding V_p fixed. This also
needs to be fixed.

I’m not sure why the authors chose to use a shortening rate instead of a constant velocity. It is
more general than the constant velocity case, but more difficult to characterize. By adding
another parameter, they need to show, and constrain, another dimension in parameter space,
so simple summarized like Figure 1 will no longer be possible. This also means that any real
applications will need further independent constraints.

Response: First of all, we appreciate the concerns raised by the reviewer about the problem in
semantics on advection. Throughout the revised manuscript, ‘advection gradient’ and
‘advection rate’ have been corrected as ‘advection velocity gradient’ and ‘advection velocity’,
respectively. Second, we agree with the reviewer that the simplest form of advection would
be to add a constant horizontal velocity to the landscape, with boundary conditions of fixed
position as well as elevation. On the other hand, as the reviewer has pointed out that the linear
advection velocity gradient, i.e., a constant shortening rate, is more general than the constant
velocity case. This is the reason why we have chosen a constant shortening rate. However, in
this case, both the velocity gradient and the velocity at one boundary should be constrained
simultaneously, because these two parameters determine the average velocity which is the
first-order factor that determines the steady-state location of the MDD. In our previous
numerical simulations and the analytical solution, we have only constrained the advection
velocity gradient.

Following the reviewer’s suggestion, in the revised manuscript we have included both the
constant advection velocity and the linear velocity gradient cases in our theoretical
considerations (see Fig. 2a below). For the constant advection velocity case, the only variable
is advection velocity. In this case, normalized steady-state divide location increases linearly
with the velocity. When the velocity is zero (i.e., no advection), the MDD forms at the centre
of the mountain belt. When the velocity reaches about 5.1 mm/yr, the normalized steady-state
divide location is 100%, namely, there is no MDD within the topography. Furthermore, the
analytical solution (see Methods) shows that the velocity, at which the normalized
steady-state divide location reaches 100%, depends on erodibility, mountain width, and
hillslope length. When there is a linear advection velocity gradient, the normalized
steady-state divide location increases with both the velocity gradient and the velocity at the
retro-wedge edge (V_r).

3. The second problem with advection is that the natural examples are still not calculating the
 advection velocity correctly. The Taiwan example is more or less correct in that they have the
 correct velocity frame of reference now. However, the Sicily example still lacks a frame of
 reference, which will determine the mean value of the advection velocity. Currently the
 authors show that the velocity field has no gradient, but they take this to imply that advection
 is therefore negligible. Noting point above, this is not valid. It is the mean velocity with
 respect to the Earth's surface (assumed to be steady), not the gradient that is important. To
 estimate the advection velocity in Sicily, as in Taiwan, it is necessary to find the velocity of
 the rock with respect to the surface. If the surface is in steady state, that means the boundaries
 at the coast are the appropriate reference frame. The authors need to give the velocity of the
 coast with respect to the rock. At the moment there is no attempt to do this, but without this
 number, the advection velocity is unknown. I would guess, given the proximity of the coast to
 the MTF, that it is nearly fixed with respect to the hanging wall of that fault, and therefore the
 advection velocity is close to the slip rate on the fault, but the authors need to do some
 analysis of this.

Response: In the previous round of revision, following the suggestion from the reviewer, we
 chose the Longitudinal Valley as the reference frame for advection in Southern Taiwan. Other
 points are moving towards the SE with respect to the Longitudinal Valley. We thank the
 reviewer not only pointed out the necessity of including advection in our model, but also gave
 us specific suggestions on how to choose the reference frame of advection to calculate the
 advection velocity.

As suggested, we chose Western Calabria as the reference frame for advection in
 Northeastern Sicily. Based on three GPS data along the Western Calabria, and eighteen GPS
 data on Northeastern Sicily, we obtained an average advection velocity of 0.77 mm/yr.
 Meanwhile, we also used the SE coastline of Northeastern Sicily as the reference frame for
 advection. Based on six GPS data along the SE coastline of Northeastern Sicily, and the rest
 twelve GPS data on Northeastern Sicily, we acquired an average advection velocity of 0.46
 196 mm/yr. Accordingly, we consider that the advection velocity is spatially uniform in
 Northeastern Sicily with value of 0.46 or 0.77 mm/yr (Fig. 4a) (please see Lines 390–404 of
 the revised manuscript for detail).

 Based on the precipitation data in Fig. 4b, we calculated the average annual precipitation rate
 of the SE and NW sides of Northeastern Sicily to be 705 and 711 mm/yr, respectively. We
 measured 34 sites within Google Earth and obtained an average hillslope length of 1756 ± 294
 203 m. We calculated the average channel steepness to be 143 m and 179 m for the NW and SE
 sides, respectively. Based on the obtained channel steepness, we estimated the erodibility
 values of $5.4 \times 10^{-6} \text{ yr}^{-1}$ and $4.9 \times 10^{-6} \text{ yr}^{-1}$ for the NE and SE sides, respectively. By
 incorporating the above parameters into MATLAB script (see Code availability), we
 estimated the divide locations under different settings (see Fig. S9 below) (please see Lines
 405–420 of the revised manuscript for detail).

With the newly-acquired advection velocity in Northeastern Sicily, we modified the main text
as: *'In Northeastern Sicily, the uplift is mainly controlled by the tectonic activity of the*
*Messina-Taormina Fault (MTF), being higher on the SE side (Fig. 4a). The estimated uplift*
*gradient of 0.0092 mm/yr/km (see Methods) pushes the MDD towards the SE. In the direction*
*perpendicular to the MDD, the average advection velocity is estimated to be 0.46 or 0.77*
*mm/yr (Fig. 4a) (see Methods). The northwestwards advection pushes the MDD towards the*
*NW. Both the mean annual precipitation (711 and 705 mm/yr for the NW and SE flanks,*
*respectively) and lithology are almost symmetrical with respect to the MDD (Fig. 4b,c), and*
*thus should have little influence on divide location. We choose the ratio between the width of*
*the NW side and the mountain width as the normalized divide location, d . Although the d - λ*
*data points generally follow the trend of the analytical solution for mountain belts with uplift*
*gradient, most points are below the analytical solution due to the existence of advection (Fig.*
*1e). We numerically solve the divide location under different geological settings including*
*both uplift gradient and advection, where no analytical solution is possible (see Code*
*availability). For an advection velocity of 0.46 mm/yr, the MDD will form at $d = 56.7\%$ under*
*the present settings, comparable with the current divide location of 58.7%, implying that the*
*tectonic activity remains constant over the response timescale of the MDD. This value only*
*slightly decreases to $d = 56.4\%$ if we exclude the precipitation difference on both sides of the*
*range. If we exclude uplift gradient and advection, the MDD is predicted at $d = 49.4\%$ and*
*62.2%, respectively (Supplementary Fig. S9). Alternatively, for an advection velocity of*
*0.77 mm/yr, the MDD should form at $d = 52.8\%$ under the present settings, lower than the*
*current divide location. We attribute this discrepancy to the decreased uplift gradient and/or*
*increased advection velocity. In summary, we expect that the tectonic activity of the*
*Northeastern Sicily remains constant or has slightly changed.'* (Lines 198–220 of the revised
manuscript).

4. There is a third problem with the treatment of advection and that is the way the divide
mobility statistics have been applied (Fig. S6). These statistics were developed for regions
undergoing uplift only, without advection and are either not valid or must be interpreted
differently for settings with advection. For a setting with steady state topography and
advection, the divide MUST move at the advection velocity, but in the other direction. Right?
Otherwise it is not steady under the advection velocity. Interpretation of the geomorphic
indicators must reflect this, although it depends somewhat on whether they are local (relief,
slope at divide) or global (chi). If indicators really all show divides are stable (they don't look
very stable to me), and there is steady state, there is no advection. Incidentally, this is also
somewhat true with uplift gradients. Particularly chi values, but also steepness index needs to
be corrected for uplift gradient.

Response: Based on the principle that divide tends to migrate to the side with lower erosion
rate (lower topographic relief and slope) (*Gilbert, 1887; Goren et al., 2014; Willett et al.,*
*2014*), *Forte and Whipple (2018)* developed a tool to estimate divide stability. The tool
measures the topographic metrics, including relief, slope, elevation, and Chi. The relief, slope,
and elevation are only measured near the divide, with a reference drainage area controlling
the width of the area near the divide. The Chi index was developed by *Willett et al., 2014*.
Under uniform uplift and precipitation settings, Chi value contrasts across the divide can be
used to estimate divide mobility. Divide tends to move to the side with higher Chi values. In
the case that uplift or precipitation is not uniform, the Chi contrast may only indicate a
possible future divide instability (*Forte and Whipple, 2018*).

We appreciate that the reviewer reminds us that these divide metrics are unable to directly
estimate divide mobility in the case with advection. For Northeastern Sicily and Southern
Taiwan with advection, we cannot use the above divide stability metrics to estimate the
stability of their MDDs. Therefore, we have removed Fig. S6 of the previous manuscript.
Besides, we have revised the related interpretation for Southern Taiwan in the main text (see
Lines 399–421 of the Response Letter for detail).

Additionally, we agree with the reviewer that uplift gradient may also influence the accuracy
of the divide metrics. The existence of uplift gradient may make Chi unable to estimate
short-term divide mobility accurately, but Chi can reveal the overall topography on two sides
of the mountain belt, which means that it may also have predictability over long timescale
(*Forte and Whipple, 2018*). The three other divide metrics (topographic relief, slope, and
elevation) are measured near the divide. The width of the analysed area depends on the value
of the reference drainage area. For example, if the reference drainage area for each side is
1 km^2 , the width of the analysed area for each side of the mountain belt is about 1 km. For a
certain uplift gradient, a larger reference drainage area means a larger uplift difference
between the two sides across the divide, which will have more influence on the accuracy of
the divide metrics. Therefore, for Wula Shan horst (Fig. S8), we used a reference drainage
area of 0.01 km^2 , indicating the width of the analysed area is just about 10 m for each side.
Within this narrow range, the uplift rate difference across the divide can be ignored. Thus, the
results of the divide metrics are reliable in this case.

5. The Hacks Law problem described starting at line 70 is pretty well known in the landscape
modeling community, at least among those of us who have worked on advection. The models
in figure 1e, and f are not realistic landscapes. The advection has led to the creation of nearly
perfectly uniform width, unbranching rivers, so the Hack Exponent on the Pro-wedge rivers is

not correct and as noted is a function of how much advection is present. This raises the
question as to what is the point of these numerical models. If they are to confirm the
analytical model assumption that Hack's Law holds, they need first to reproduce Hack's Law.
Otherwise I don't see the purpose of these models any longer.

**Response:** Previously, we aimed to compare the 11 numerical simulations with constant
shortening rates to the analytical solution. As stated by the reviewer, the pro-wedge drainage
basins are long and narrow, which can hardly be observed in natural landscapes, and thus may
not be realistic landscapes. Therefore, we have removed the results of the 11 numerical
simulations (see also in Lines 84–92 of the Response Letter).

**Detailed comments by line number:**

6. Line 30-45: I'm not convinced that cost effectiveness is a very good selling point for this
analysis. Divide position depends on uplift gradient, precip gradient, advection velocity and
lithology, so the problem is multidimensional already and most of these methods need to be
applied to constrain the problem anyway. Why not sell it on just learning something more –
applying another constraint for internal self-consistency? Something fundamental. It would
feel less contrived.

**Response:** After fixing the problem of advection in the analytical solution, we have
reorganized the selling point of this paper. In the revised Introduction, instead of the cost
effectiveness of our model, we emphasize the value of quantifying the relationships between
mountain asymmetry, determined by the location of the MDD, and both uplift gradient and
advection velocity and its gradient.

Specifically, we rewrote the last paragraph of the Introduction as:

*'Here we present a new method to provide constraints on the tectonic pattern and history of a*
*mountain belt solely based on the location and mobility of its main drainage divide (MDD).*

*First, we quantify the relationship between mountain asymmetry, determined by MDD*
*location, and both uplift gradient and advection velocity by theoretical reasoning and*
*landscape evolution modelling. Then, we illustrate the interaction and competition between*
*tectonic deformation and erosion in determining MDD mobility. Collectively, we build a*
*model to constrain the present or previous tectonic information of a mountain belt just based*
*on the current location and mobility of its MDD, followed by the applications to Wula Shan*
*horst, Northeastern Sicily, and Southern Taiwan.'* (Lines 39–47 of the revised manuscript).

7. Line 70. There is something interesting going on with the hack coefficients and exponents,
but what? Why are they different? It would help to break out the Pro-wedge and retro-wedge
rivers separately to see more.

Response: We agree to the suggestion that it would help to separate the topography. Because
we have removed the 11 numerical simulations with advection velocity gradient, we only
separated the topographies produced by the 20 numerical simulations with different uplift
gradients. We focus on the top (with higher uplift rates) and bottom sides, where rivers flow
to the top and bottom edges, respectively. The left and right sides are excluded from the
analysis. We analyzed the data from the top and bottom sides to fit the Hack's Law,
respectively (Figs. S2 and S3). Then, we plotted the relationship between Hack's parameters
and the uplift gradients in Fig. S4. Specifically, for the top side, the Hack's exponent (b) is
relatively stable when the uplift gradient is small (less than 0.13 mm/yr/km). Then, in general,
the exponent increases with uplift gradient. For the bottom side, the exponent is relatively
stable for most of the uplift gradients, except for the lowest and highest uplift gradients. It
seems that Hack's parameters are not always constant with varying uplift gradients. However,
overall, there is no systematic relationship between the Hack's parameters and the uplift
gradients. Therefore, at the moment, we are still unable to explain the discrepancy between
the analytical solution and the results of numerical simulations exactly. This would be an
interesting topic for future research.

8. Line 117. I think other studies, including ref 5, show the opposite. Adding advection
increases time to SS.

Response: Similar to ref. 5 (*Willett et al., 2001*), our results show that the topography of a
mountain belt with horizontal advection is hard to reach a steady state. The mean elevation is
not stable even at an overall steady state (Fig. S4 of the previous manuscript). Since we have
removed the results of 11 numerical simulations with advection velocity gradient, the
statement in Line 177 of the previous manuscript has also been removed.

9. Line 149. See above. This is not advection velocity. This is shortening rate or rock velocity
with respect to Europe. This needs to be put in the reference frame of the land surface, as was
done with Taiwan. To do this, you must find a point where the land surface is not moving
with respect to the Eulerian surface that defines the land surface. Advection is complicated.

Response: We agree with the reviewer that advection is complicated. In the previous
manuscript, we used the uniform GPS horizontal velocity to infer that the advection in
Northeastern Sicily is negligible. The problem was that we did not have a right reference
frame for advection. Based on the suggestion from the reviewer that the advection velocity is
close to the horizontal slip rate on the MTF, we chose Western Calabria, on the hanging wall
of the MTF, as the reference frame for advection. In the direction perpendicular to the MDD,
the average advection velocity is 0.77 mm/yr. Meanwhile, in the previous round review report,
Dr. Willett suggested that advection could also be defined as the rock velocity with respect to

the erosional boundary. Therefore, we also used the SE coastline of Northeastern Sicily as the
reference frame for advection. In this reference frame, the average advection velocity in the
direction perpendicular to the MDD is 0.46 mm/yr. With these two advection velocities, the
MATLAB script, and some other new data, such as Hack's parameters, average precipitation,
hillslope length, and erodibility, we re-analysed the Northeastern Sicily case (please see Lines
190–236 of the Response Letter for details).

10. Line 177. Just one example of problem throughout, but it is the advection velocity that
pushed the divide, not the advection gradient.

Response: We appreciate this comment. We have revised 'advection gradient' and 'advection
rate' as 'advection velocity gradient' and 'advection velocity', respectively, throughout the
main text, Supplementary Information, and Source Data file.

11. Line 188. Erosion rates from ^{10}Be are averaging over less than 10ka. For the GPS uplift
data to be consistent means that tectonic uplift rates have increased by 2 to 10 times over a
few thousand years over all of south Taiwan. This is not realistic. I think it is likely that the
GPS are either wrong (vertical rates are hard to measure and signals always come down as
more data are collected. Or they are dominated by a large interseismic elastic accumulation.

Response: According to the reviewer's suggestion, we compared ^{10}Be -derived erosion rates
with GPS-derived uplift rates in detail. On the SE side of our study area, the average erosion
rate of one drainage basin (Hsinwulu River) was estimated to be 4.5 mm/yr (*Fellin et al.,*
*2017*), which is in contrast to the GPS-derived average uplift rate of ~ 9.4 mm/yr in the same
basin. However, the timescale of just about 130 years for the erosion rate (*Fellin et al., 2017*)
is comparable to that of the GPS observations (2002–2013), which, consequently, cannot
provide robust evidence for temporal variation in tectonic uplift as we interpreted in Lines
188–190 of the previous manuscript.

We agree that uplift rates and advection velocity estimated from GPS observations over
several decades may have been over-estimated, because the study area displayed a high level
of seismicity (Chen et al., 2016) and hence is dominated by a large interseismic elastic
accumulation. Therefore, we removed the related statement in the revised manuscript.
Meanwhile, we re-analysed the discrepancy between the current divide location of $d = 64.7\%$
and the predicted value of $d = 78.1\%$ (see Lines 410–421 of the Response Letter for detail).

12. Line 190. Topography cannot be steady over a short timescale and unsteady over a long
timescale. This doesn't make sense. In addition the numerical models showed that steady state
requires tens to hundreds of Ma to reach steady state, this hypothesized transient is occurring
over a few thousand years. Cannot possibly be in steady state with the divide position
reflected the GPS.

Response: Previously, we inferred the topography in a steady state from the divide stability
metrics (Chi, relief, slope, and elevation) that predict the MDD of Southern Taiwan to be
stable (Fig. S6 in the previous Supplementary Information). However, we agree to the
statement in the reviewer's fourth comment that '*These metrics were developed for regions*
*undergoing uplift only. For a setting with steady state topography and advection, the divide*
*must move at the advection velocity, but in the other direction.*'. Both the previous work
(Willett et al., 2001) and this study pointed out that there is a strong advection in Southern
Taiwan, from NW to SE, suggesting that the current divide mobility cannot be estimated by
the divide stability metrics. On the contrary, the co-existence of advection and stable divide
(predicted by the metrics) supports the interpretation that the topography is not in steady state.

Accordingly, we have modified our interpretation as:
'*We attribute the discrepancy between the current divide location of $d = 64.7\%$ and the*
*predicted value of $d = 78.1\%$ to two possible reasons. One is that the uplift rates and*
*advection velocity estimated from GPS observations over several decades may have been*
*over-estimated, because the study area displayed a high level of seismicity (Chen et al., 2016)*
*and hence is dominated by a large interseismic elastic accumulation. The other possible*
*reason is a recent increase in tectonic activity, which is likely due to the southward*
*propagation of the arc-continent collision in Taiwan (Suppe, 1981; Willett and Brandon,*
*2002; Willett et al., 2003; Fellin et al., 2017). Under such a condition, it implies that the*
*mountain belt is in a transient state with respect to its divide position. We can expect that*
*under persistent conditions, the MDD of Southern Taiwan will migrate southeastwards,*
*approaching a divide location at $d = 78.1\%$.*' (Lines 254–263 of the revised manuscript).

13. Figures 4,5,6, : this is a strong and complicated emphasis on transients, when it is not
clear how this will be constrained. With the multiple dimensionality of the problem, it is clear
why it has become complicated, but not clear how this will be resolved. Much data, that
generally does not exist would be needed and given that uplift, advection and rainfall are
independent, it is not clear that real life cases will ever fall into these categories. Furthermore,
there is no known way of establishing divide stability in the cases with advection. As
discussed above, published divide metrics are only valid for cases without advection. I would
recommend that this entire discussion be shortened and simplified.

Response: The mobility and steady-state location of the MDD are influenced by uplift
gradient, advection, precipitation, and lithology. The quantitative relationships between
normalized divide location and tectonic forcing (uplift gradient, advection velocity and its
gradient) (Figs. 1,2), and the conceptual framework built in this study (Fig. 3) help us to
provide first-order constraints on tectonic information from the current location and stability
of the MDD.

If the controls of precipitation, lithology, and advection on divide location can be ruled out as
dominant, we can distinguish five possible cases. For a symmetric mountain belt, the uplift is
uniform if the MDD is stable (Fig. 3c). When the MDD is migrating towards the top side, the
uplift is now higher on the top side, and was uniform or higher on the bottom side (Fig. 3d).

For an asymmetric mountain belt, we can apply the theoretical relationship (Fig. 1e) to
acquire a reference uplift gradient λ' from the current divide location. If the MDD is stable,
the uplift is higher on the steeper side with a gradient of λ' (Fig. 3e; like the Wula Shan horst
example). When the MDD of an asymmetric mountain belt is unstable, λ' can provide a lower
bound on the present or previous gradient. If the MDD is migrating towards the steeper side,
the uplift is higher on the steeper side with a gradient greater than λ' (Fig. 3f). When the
MDD is moving to the gentler-sloping (bottom) side, the uplift was higher on the top side
with a gradient greater than λ' (Fig. 3g). Additionally, if the present uplift is known to be
higher on the top side, the uplift gradient should have decreased.

In principle, the current divide location and migration pattern can also be used to constrain
advection pattern (Fig. S7). However, there is no known way of establishing divide stability
in the cases with advection. Thus, our model is unable to constrain advection information
quantitatively until a novel method that could establish divide mobility under advection is
developed in the future. Nevertheless, if the present advection velocity is known, our model
together with our MATLAB script can provide information on the previous tectonic
deformation, depending on the specific uplift and advection pattern (see Northeastern Sicily
and Southern Taiwan example in Discussion section, Lines 166–172 of the revised
manuscript).

For the clarity of the manuscript, we have incorporated the above descriptions and analyses
into the Discussion section of the revised manuscript.

Meanwhile, for simplicity, we have merged the previous Figs. 4 and 5 in the revised
manuscript (new Fig. 3). Besides, Fig. 7 of the previous manuscript has been moved into
Supplementary Information as Fig. S11.

14. Line 430: Willett et al., Nature, 2018 gives the advection solutions for general n , m , b , but
no uplift.

Response: In Willett et al., Nature, 2018, the advection term was also considered in solving
the problem of the migration rate of drainage divide. The applied reference frame is the same
as Goren et al., 2014 and our work, i.e., $x = 0$ at the MDD. Besides, they also studied the
relationship between landscape evolution and hydrological systems. Thus, we have cited this
paper in the revised manuscript as:

1. Mountain belts determining regional weather and the distribution of hydrological systems
(Hoorn et al., 2010; Willett et al., 2018) (Line 24 of the revised manuscript).

2. The advection theory is based on Goren *et al.*, 2014 and Willett *et al.*, 2018 (Line 334 of
the revised manuscript).

**References cited in Response Letter**

Chen, Y. L., Hung, S. H., Jiang, J. S., & Chiao, L. Y. Systematic correlations of the earthquake
frequency-magnitude distribution with the deformation and mechanical regimes in the
taiwan orogen. *Geophys. Res. Lett.* **43**, 5017–5025 (2016).

Fellin, M. G., Chen, C. Y., Willett, S. D., Christl, M., & Chen, Y. G. Erosion rates across
space and timescales from a multi-proxy study of rivers of eastern Taiwan. *Global Planet.*
*Change* **157**, 174–193 (2017).

Forte, A. M., & Whipple, K. X. Criteria and tools for determining drainage divide stability.
*Earth Planet. Sci. Lett.* **493**, 102–117 (2018).

Gilbert, G.K. Geology of the Henry Mountains (Utah). USGS Report, Government
Printing Office, Washington, D.C (1877).

Goren, L., Willett, S. D., Herman, F., & Braun, J. Coupled numerical-analytical approach to
landscape evolution modeling. *Earth Surf. Process. Landf.* **39**, 522–545 (2014).

Hoorn, C., et al. Amazonia through time: Andean uplift, climate change, landscape evolution,
and biodiversity. *Science* **330**, 927–931 (2010).

Suppe, J. Mechanics of mountain building and metamorphism in Taiwan. *Mem. Geol. Soc.*
*China* **4**, 67–89 (1981).

Whipple, K.X., Forte, A.M., DiBiase, R.A., Gasparini, N.M., & Ouimet, W.B. Timescales of
landscape response to divide migration and drainage capture: Implications for the role of
divide mobility in landscape evolution. *J. Geophys. Res. Earth Surf.* **122**, 248–273
(2017).

Willett, S.D., Slingerland, R., & Hovius, N. Uplift, shortening, and steady state topography in
active mountain belts. *Am. J. Sci.* **301**, 455–485 (2001).

Willett, S. D., & Brandon, M. T. On steady states in mountain belts. *Geology* **30**, 175–178
(2002).

Willett, S. D., Fisher, D., Fuller, C., En-Chao, Y., & Chia-Yu, L. Erosion rates and
orogenic-wedge kinematics in Taiwan inferred from fission-track thermochronometry.
*Geology* **31**, 945–948 (2003).

Willett, S. D., McCoy, S. W., Perron, J. T., Goren, L., & Chen, C. Y. Dynamic reorganization
of river basins. *Science* **343**, 1248765 (2014).

Willett, S. D., McCoy, S. W., & Beeson, H. W. Transience of the North American High Plains
landscape and its impact on surface water. *Nature* **561**, 528–532 (2018).

REVIEWERS' COMMENTS

Reviewer #2 (Remarks to the Author):

I've gone through the paper again and find it acceptable. Although I may not agree with all the analysis, e.g. uplift rates in Taiwan, the authors have provided enough caveats and sensitivity analysis that makes it clear where their assumptions might still be on tenuous ground, and I don't see any reason to delay publication further.

I do have a couple more comments that I would recommend the authors consider before publication. I do not need to see the paper again.

(1) I think you should consider an addition to figure 1, that shows in cartoon form, how you are treating a mountain belt. You have now largely adopted a wedge framework (my fault probably for pushing this), for example using pro-wedge and retro-wedge terminology, but I expect readers will not get this without some explanation. The entire concept of advection is not readily intuitive. In addition, Sicily and the Wula Graben/Horst system are not wedges, so you need to explain the kinematics here and not use the wedge terminology. I would add a figure defining advection and uplift in each of these systems. Something like I sketch here below. Particularly for the broader readership of the journal.

(2) Line 87. There are many more than 2 situations for mountain belt kinematics. There are two simple parameterizations, that you consider here. State it like that.

(3) Line 105. Sicily is not a wedge setting. Make this clear. How does advection come into play with a normal fault. See figure 1 comment.

(4) Line 129. Erosion is not a parameter. It is an outcome of the model. Also, you mean erosion rate.

(5) Several places. The divide stability analysis of Forte and Whipple that is used, I think you didn't get my point last time. The divide analysis is not for stability. It is for asymmetry. It provides a set of metrics to characterize divide asymmetry with the idea that only symmetric divides are stable. This is true only for constant uplift. If uplift has a gradient, or in the presence of advection, only ASYMMETRIC divides are stable. Symmetric divides are unstable, so the printed conclusion on top of the Forte plots is wrong. You are not using this analysis correctly. I would change this or take it out.

**Response Letter**

We thank the reviewer for the further suggestions, which greatly helped us to improve the
clarity of this manuscript. We address these concerns point by point, and highlight
implemented changes in the manuscript.

**Responses to Reviewer #2 (Sean Willett)**

I've gone through the paper again and find it acceptable. Although I may not agree with
all the analysis, e.g. uplift rates in Taiwan, the authors have provided enough caveats
and sensitivity analysis that makes it clear where their assumptions might still be on
tenuous ground, and I don't see any reason to delay publication further.

I do have a couple more comments that I would recommend the authors consider
before publication. I do not need to see the paper again.

**Response: We thank Reviewer #2 for positive comments and the recommendation of final**
**publication. His insightful suggestions during the revisions are highly appreciated as well.**

(1) I think you should consider an addition to figure 1, that shows in cartoon form, how
you are treating a mountain belt. You have now largely adopted a wedge framework
(my fault probably for pushing this), for example using pro-wedge and retro-wedge
terminology, but I expect readers will not get this without some explanation. The entire
concept of advection is not readily intuitive. In addition, Sicily and the Wula
Graben/Horst system are not wedges, so you need to explain the kinematics here and
not use the wedge terminology. I would add a figure defining advection and uplift in
each of these systems. Something like I sketch here below. Particularly for the broader
readership of the journal.

**Response: We agree with the reviewer that a cartoon explaining how we are treating a**
**mountain belt is necessary and can improve the readability of this paper. Based on the sketch**
**provided by the reviewer, we have added an extra figure (new Fig. 1 as below) in the revised**
**manuscript, defining tectonic uplift and horizontal advection in both contraction and**
**extension systems.**

 **Fig. 1 Kinematic models for tectonic uplift and horizontal advection.** Material transport
 (marked as dashed lines with arrows), with vertical component (V_y) and horizontal
 components (V_x), causes tectonic uplift and horizontal advection in convergent orogen a
 (modified after Willett *et al.*, 2001; Willett and Brandon, 2002) and extensional orogen b. For
 each system, advection is from positive side to negative side. For the positive side, due to
 advection, the range half-width tends to increase. By contrast, the range half-width of the
 negative side tends to decrease. In contraction system, the positive and negative sides are
 pro-wedge and retro-wedge, respectively.

 (2) Line 87. There are many more than 2 situations for mountain belt kinematics. There
 are two simple parameterizations, that you consider here. State it like that.

**Response:** We agree to the reviewer's comment and have changed the sentence into:
 *'Generally, there are multiple possible situations for mountain belt kinematics. Here, we*
 *consider two simple parameterizations of mountain belt kinematics with advection.'*, as
 suggested by the reviewer.

 (3) Line 105. Sicily is not a wedge setting. Make this clear. How does advection come
 into play with a normal fault. See figure 1 comment.

**Response:** We agree with the reviewer. To avoid misunderstanding, the two sides of a
 mountain belt in both contraction and extension systems are named after 'positive side' and
 'negative side', respectively (Fig. 1 of the revised manuscript). Meanwhile, we have corrected
 the related terminology throughout the main text and SI.

 (4) Line 129. Erosion is not a parameter. It is an outcome of the model. Also, you mean
 **Response:** We agree that the erosion rate is not a parameter in numerical simulation. The
 difference in cross-divide erosion rate tends to push the divide to move. We have revised the
 sentence as:

*'Divide dynamics can be understood conceptually by focusing on three forces: advection,*
*asymmetric uplift, and asymmetric erosion.'*

(5) Several places. The divide stability analysis of Forte and Whipple that is used, I think
you didn't get my point last time. The divide analysis is not for stability. It is for
asymmetry. It provides a set of metrics to characterize divide asymmetry with the idea
that only symmetric divides are stable. This is true only for constant uplift. If uplift has a
gradient, or in the presence of advection, only ASYMMETRIC divides are stable.
Symmetric divides are unstable, so the printed conclusion on top of the Forte plots is
wrong. You are not using this analysis correctly. I would change this or take it out.

*Response: In the previous round of review, the reviewer reminded us that the divide metrics*
*are unable to directly estimate divide mobility in the case with advection. Therefore, we*
*removed the divide metrics in Taiwan and Sicily examples. We also agree that uplift gradient*
*may also influence the accuracy of the divide metrics. Uplift gradient may make Chi unable*
*to estimate short-term divide mobility accurately, but Chi has predictability over long*
*timescale. The other three divide metrics (topographic relief, slope, and elevation) are*
*measured near the divide. The width of the analysed area depends on the value of the*
*reference drainage area. For a given uplift gradient, the larger reference drainage area we*
*choose, the larger difference in uplift rate between the two sides across the divide exists. For*
*example, if the reference drainage area is 1 km², the width of the analysed area for each side*
*of the mountain belt is about 1 km, which may have nonnegligible impacts on the accuracy of*
*the divide metrics. However, for Wula Shan horst (Fig. S8), we used a reference drainage area*
*of 0.01 km², i.e., the width of the analysed area of just about 100 m for each side. Within this*
*narrow range, the difference in uplift rate across the divide is negligible. Thus, the results of*
*the divide metrics are reliable in this case.*

**References**

Willett, S.D., Slingerland, R., & Hovius, N. Uplift, shortening, and steady state topography in
active mountain belts. *Am. J. Sci.* **301**, 455 – 485 (2001).

Willett, S. D., & Brandon, M. T. On steady states in mountain belts. *Geology* **30**, 175–178
(2002).